# Laser activation of single group-IV colour centres in diamond

Xingrui Cheng [1,2], Andreas Thurn [2,3], Guangzhao Chen[1,7], Gareth S. Jones[1], James E. Bennett [1], Maddison Coke[4], Mason Adshead[4,5], Cathryn P. Michaels[3], Osman Balci [6], Andrea C. Ferrari[6], Mete Atatüre [3], Richard J. Curry [4,5], Jason M. Smith [1] ✉, Patrick S. Salter [2] ✉ & Dorian A. Gangloff [2,3] ✉

Spin-photon interfaces based on group-IV colour centres in diamond offer a promising platform for quantum networks. A key challenge in the field is realising precise single-defect positioning and activation, which is crucial for scalable device fabrication. Here we address this problem by demonstrating a two-step fabrication method for tin vacancy (SnV⁻) centres that uses site-controlled ion implantation followed by local femtosecond laser annealing with in-situ spectral monitoring. The ion implantation is performed with sub-50 nm resolution and a dosage that is controlled from hundreds of ions down to single ions per site, limited by Poissonian statistics. Using this approach, we successfully demonstrate site-selective creation and modification of single SnV⁻ centres. Our in-situ spectral monitoring opens a window onto materials tuning at the single defect level, and provides new insight into defect structures and dynamics during the annealing process. While demonstrated for SnV⁻ centres, this versatile approach can be readily generalised to other implanted colour centres in diamond and wide-bandgap materials.

Colour centres in diamond have garnered significant attention for their use in quantum technologies[1,2] such as quantum simulators[3], quantum sensors[4] and quantum networking interfaces[5,6]. Among these, nitrogen-vacancy (NV⁻) centres are the most extensively studied due to their ground-state spin's long coherence times at room temperature[7,8]. However, their relatively low emission fraction into a purely photonic mode—the zero-phonon line (ZPL)[9] – and susceptibility to spectral diffusion, particularly near surfaces[10,11], poses a significant challenge for their use in optical quantum technologies. In contrast, group-IV colour centres have emerged as potential alternatives due to their crystallographic inversion symmetry, which leads to dominant ZPL emission[12], manageable spectral diffusion near surfaces in nanophotonic devices[13], and reduced inhomogeneous broadening[14,15], while preserving sufficient ground-state spin coherence at cryogenic temperatures[16]. Tin-vacancy (SnV⁻) centres stand out among group-IV defects due to their optimal spin-orbit coupling, which is significantly larger than for silicon-vacancy (SiV⁻)[17] and germanium-vacancy (GeV⁻)[18] centres, protecting spin coherence against phonon scattering under standard cryogenic conditions (~2 K)[10], but smaller than for lead-vacancy (PbV⁻) centres, allowing magnetically driven ground-state spin control with moderate strain levels[19]. Despite these promising attributes, a method to create and activate single optically-active SnV⁻ (and other group-IV) centres that combines high spatial accuracy and high efficiency remains an outstanding challenge to scale up quantum technologies with these emitters.

[1]Department of Materials, University of Oxford, Parks Road, Oxford OX1 3PH, UK. [2]Department of Engineering Science, University of Oxford, Parks Road, Oxford OX1 3PJ, UK. [3]Cavendish Laboratory, University of Cambridge, J. J. Thomson Avenue, Cambridge CB3 0HE, UK. [4]Photon Science Institute, Faculty of Science and Engineering, University of Manchester, Manchester M13 9PL, UK. [5]Department of Electrical and Electronic Engineering, Faculty of Science and Engineering, University of Manchester, Manchester M13 9PL, UK. [6]Department of Engineering, University of Cambridge, Trumpington Street, Cambridge CB2 1PZ, UK. [7]Present address: Accelerator Technology and Applied Physics Division, Lawrence Berkeley National Laboratory, Berkeley, California 94720, USA. ✉e-mail: jason.smith@materials.ox.ac.uk; patrick.salter@eng.ox.ac.uk; dag50@cam.ac.uk

Unlike nitrogen, group-IV elements are not naturally abundant in diamond. As a result, a number of different approaches have been developed to create group-IV colour centres, each with its own strengths and limitations. Chemical vapour deposition (CVD) growth[14,20,21] and high-pressure, high-temperature (HPHT) synthesis[22–24] in the presence of group-IV precursors can produce high quality emitters. However, these methods lack the ability to position individual colour centres, which is crucial for many applications. In contrast, ion implantation enables high-resolution spatial positioning while allowing for fine control over the implantation dose[25], even facilitating the deterministic placement of individual ions[26,27]. However, particularly for large atoms like tin, this method often results in lattice damage that can degrade the optical and spin properties of the created emitters[12,21,27]. Post-implantation thermal annealing, crucial for creating the colour centres and repairing lattice damage, can be performed in two distinct ways. Low-pressure, low-temperature annealing at up to 1200 °C is compatible with most nanofabrication processes but often results in incomplete damage repair, resulting in significant inhomogeneous broadening of the ZPL distribution[10,28–30]. On the other hand, high-pressure, high-temperature (HPHT) annealing, at up to 2100 °C, can effectively reduce inhomogeneous broadening and produce colour centres with superior optical properties[30,31]. However, this process often causes significant damage to the diamond surface, making it incompatible with many nanofabrication techniques. These limitations have motivated the development of alternative approaches, such as shallow ion implantation and growth (SIIG)[32]. SIIG combines low-energy ion implantation with subsequent CVD overgrowth, minimising lattice damage and thus allowing low-pressure, low-temperature annealing to sufficiently heal the diamond crystal structure[32]. This method has shown promising results in creating high-quality, site-controlled SnV⁻ centres with low inhomogeneous broadening[32]. However, SIIG—as well as the other approaches mentioned above – still face challenges, including low formation yields (1–5%)[32,33] and lack of site-specific control of the annealing process, which hinder scalable device fabrication and precise control over individual colour centre formation. These challenges are compounded by a lack of understanding of the defect structures formed by the implantation process, and the subsequent physical mechanism by which colour centre activation occurs during annealing.

Laser writing has emerged as a favourable technique for precision engineering of colour centres in crystals through vacancy generation[34–37] and laser annealing[38,39], and has made possible the deterministic creation of NV⁻ centres in diamond through live fluorescence monitoring and feedback[38]. Recent investigations have also shown that laser annealing can enhance the creation yield of SiV⁻ centres during thermal annealing[40]. These findings suggest the applicability of laser-based techniques to a broader set of emitters in diamond. Moreover, spectral analysis of single emitter fluorescence, as compatible with the laser annealing approach, could reveal valuable information on lattice defect mobilisation and the dynamics of colour centre formation, which remains an outstanding challenge.

Here we report the laser-processing of ion-implanted electronic-grade diamond to create single SnV⁻ emitters at precise positions. Our fabrication path combines site-selective ion implantation, featuring 50 nm resolution (with sub-20 nm possible)[25], with subsequent femtosecond laser annealing and live photoluminescence (PL) monitoring. By monitoring the emission spectrum during the annealing process, we observe switching between SnV⁻ and another distinct Sn-related defect that had been observed previously but not directly associated with SnV⁻[31,41,42]. This defect—which we refer to as 'Type II Sn' and hypothesise to be a SnV⁻ defect bound to a carbon self-interstitial (SnV-$C_i$)—is observed to be dominant post laser activation and appears to serve as a precursor state before a stable SnV⁻ is accessed. Live spectral evolution data during the annealing process reveals reversible transitions between the Type II Sn centres, SnV⁻ centres, and optically inactive states, providing a window into the defect formation process and making possible deterministic activation by PL-based feedback.

## Results

### Laser activation of defects in ion-implanted diamond

An illustration of the implantation and laser annealing process is shown in Fig. 1a. Doubly-ionised ¹¹⁷Sn atoms were implanted into a CVD-grown electronic-grade type IIa diamond substrate (with a nitrogen density <1 ppb) using a focused ion beam[25] with an acceleration energy of 50 keV and a calculated average penetration depth of ~20 nm (see Methods). Multiple rectangular $100 \,\mu m \times 100 \,\mu m$ arrays of implantation sites with $0.78 \,\mu m$ spacing were fabricated. Arrays were implanted with a Poissonian mean number of 1000, 500, 100, 50, 10, 5, and 1 Sn ion(s) per implantation site.

Following implantation, PL maps were obtained prior to femtosecond laser treatment by scanning a 1 mW, 532 nm continuous-wave excitation laser over the regions of interest (see Methods section). Figure 1a (left panel) shows a typical post-implantation PL intensity map, in which emission was collected in the wavelength range from 550 nm to 800 nm, revealing no visible fluorescence from the implantation sites. An absence of visible post-implantation fluorescence was observed even within the highest dosage (1000 ions/site) implanted region. A typical spectrum post-ion implantation is shown in Fig. 1b (top panel), where the only notable feature is second-order Raman scattering of the excitation laser.

Laser annealing was performed with 400 fs pulses at a wavelength of 520 nm and a repetition rate of 1 MHz, focused at the surface of the diamond. For such a repetition rate, cumulative heating effects are not expected (see Supplementary Note 1). As a preliminary activation step, the diamond was subjected to a raster scan of the activation laser across a $6 \,\mu m \times 6 \,\mu m$ region within an implanted array (covering approximately 60 implantation sites) for each implantation dose, with a dwell time of about 1 s per site, at a laser fluence of 2 J cm⁻² (see Methods section). PL imaging following this preliminary step reveals fluorescent emission from the implantation sites, as shown in Fig. 1a (right panel). We note that the preliminary laser treatment is well localised to the diffraction-limited laser spot (diameter 330 nm), as seen in Fig. 1c.

Image analysis confirms the positioning accuracy of the sites to be sub-50 nm (see Supplementary Note 2). Spectral analysis of the observed emission is presented in Fig. 1b (middle panel) and reveals three predominant features absent in the spectra immediately after ion implantation. The first is an emission peak centred at 595 nm, which is known to be associated with Sn-related defects in low-temperature annealed diamond[31] and which we refer to here as Type II Sn. The second emission peak, centred at 620 nm, is characteristic of the SnV⁻ defect, and a third peak at 740 nm is attributable to the well-known neutral carbon vacancy (GR1)[43]. To isolate the activation of SnV⁻ centres, we recorded PL images collecting only emission between wavelengths of 615 nm and 625 nm[10]. Figure 1c shows PL intensity maps within this SnV⁻ window for a range of implantation densities, revealing occasional, weak fluorescence from sites with 50 implanted ions extending to stronger fluorescence, from all sites, with 1000 implanted ions.

The SnV⁻-region of the spectrum shown in Fig. 1b (middle panel) resembles the inhomogeneously broadened spectrum of an ensemble of SnV⁻ centres formed by thermal annealing at around 800 °C (see SI of Iwasaki et al.[31]), suggesting only partial annealing of the diamond lattice with this preliminary laser treatment. To address this, we performed an extended laser treatment over a 2 h period with a laser fluence of 1.2 J cm⁻² at a single point in the diamond near to the activated array to ensure that no additional damage was created at the implantation sites of interest (see Methods section)—note that this long exposure time reduces to only an effective 3 milliseconds of actual exposure with 7.2 J of energy deposited. This extended laser

treatment was found to cause effects delocalised to several micro-metres beyond the diffraction-limited laser spot (see Supplementary Note 3, including a supplementary study of the effect of exposure duration and laser pulse energy), and resulted in noticeable narrowing and strengthening of the fluorescence peaks, as shown in Fig. 1b (bottom panel). Figure 1d, e, and f plot the relationships between

implantation dosage and the PL intensities after extended laser annealing, summed across the treated array sites, for spectral windows corresponding to Type II Sn, SnV⁻, and GR1 defects (as labelled in Fig. 1b). As anticipated, increasing implantation dosage results in increasing PL emission intensities across all three spectral windows. Interestingly, however, only the GR1 fluorescence intensity grows

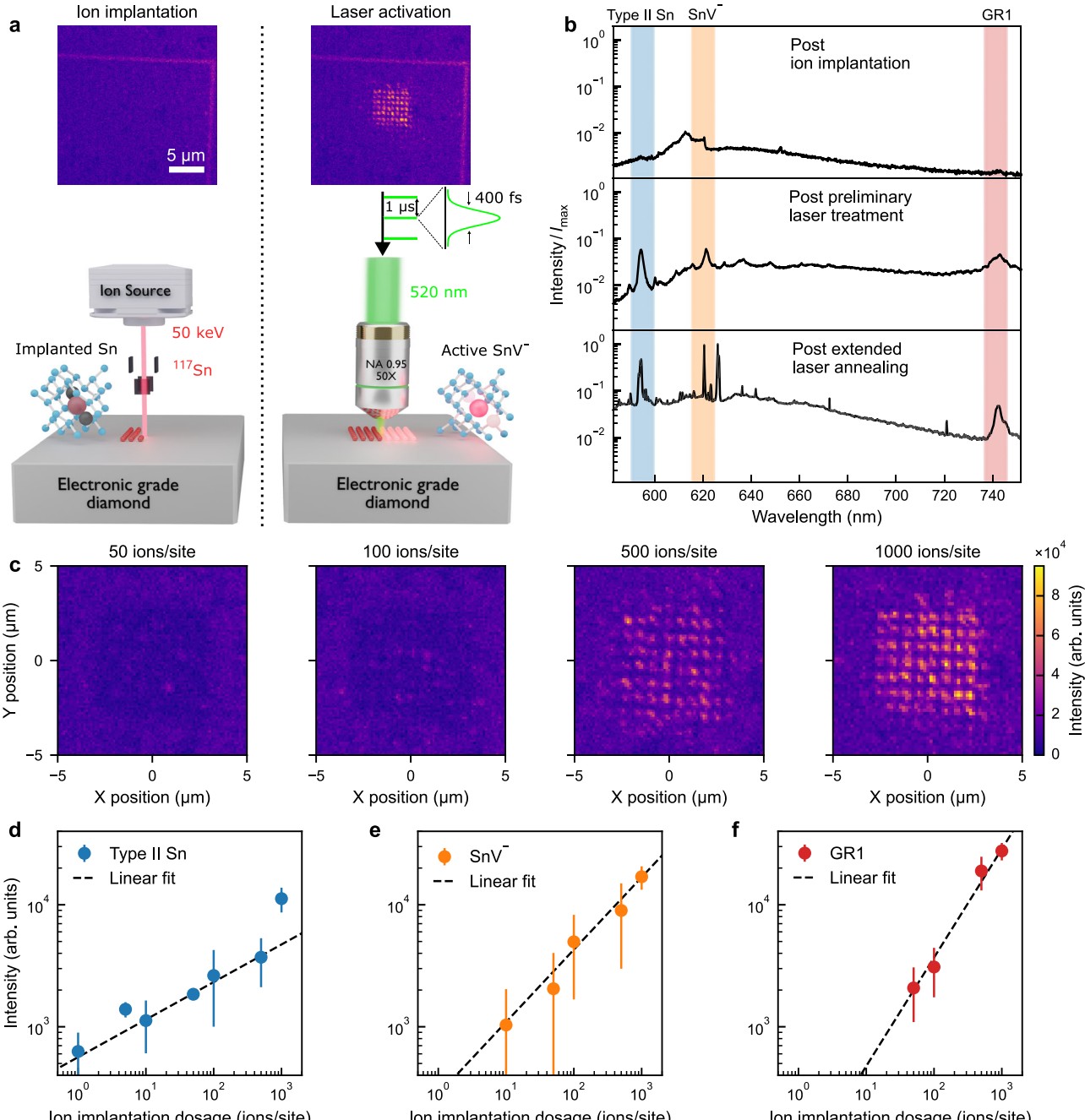

**Fig. 1 | Laser activation of tin-related defect centres. a** Schematic illustrating the process of ion implantation followed by femtosecond laser annealing. Tin (Sn$^{117}$) ions are implanted into the diamond lattice and, as a side effect, create lattice damage in the form of carbon vacancies and self-interstitials. Subsequent laser treatment activates negatively-charged tin vacancies (SnV⁻) and other defect centres. The two insets show photoluminescence (PL) images with fluorescent alignment markers before and after preliminary laser treatment, respectively. The measurement was performed at room temperature using only a 532 nm notch filter to block the PL excitation laser. **b** Example spectra from an implantation site with 500 ions at three different stages: immediately after ion implantation (top), after

preliminary laser treatment (middle), and after 2 h of extended laser annealing (bottom). Highlighted areas are the spectral windows of interest for Type II Sn (blue), SnV⁻ (orange), and neutral carbon vacancies (GR1, red). $I_{max}$ is the peak spectral intensity of the bottom spectrum. **c** PL images of four regions with different ion implantation dosages after preliminary laser treatment. The spectral collection window is 615–625 nm. **d–f** Spectrally integrated emission intensity of Type II Sn-related, SnV⁻-related, and GR1-related defects, respectively, as a function of implantation dosage following the extended laser annealing. Error bars indicate one standard error. Dashed lines are linear fits to each data set on the double logarithmic scale with slopes $0.31 \pm 0.05$, $0.59 \pm 0.27$, and $0.89 \pm 0.15$, respectively.

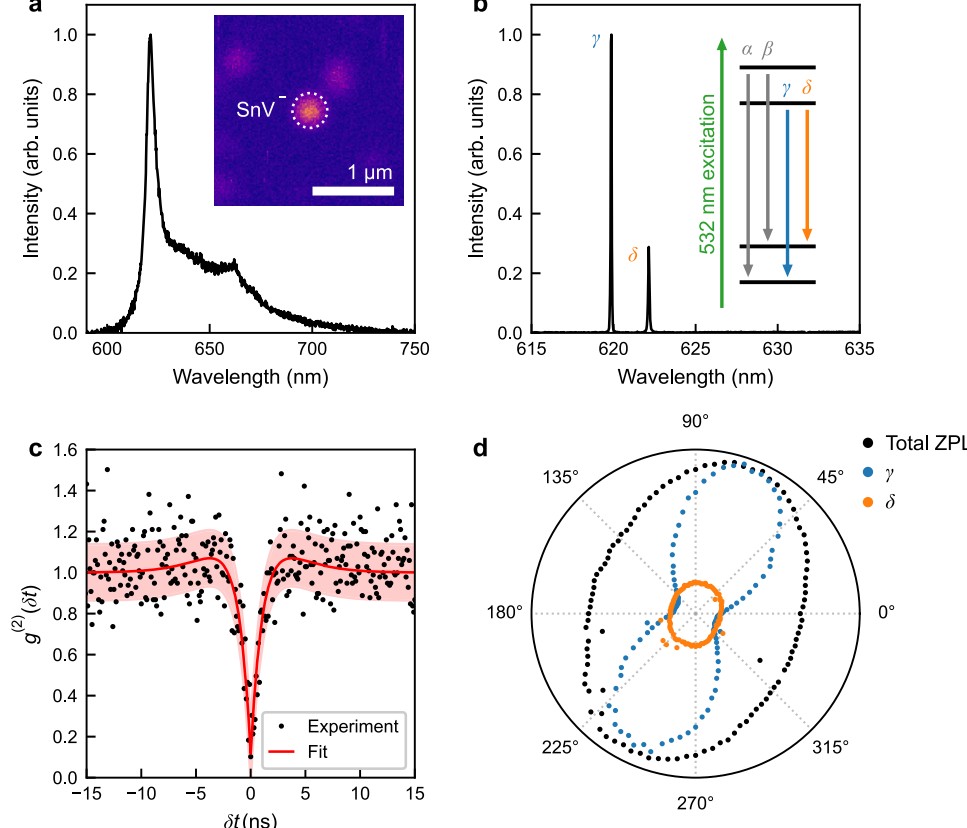

**Fig. 2 | Characterisation of a typical laser-activated single tin vacancy (SnV⁻) centre. a** Room-temperature photoluminescence (PL) spectrum from a single site in the array implanted with 10 ions per site, exhibiting the characteristic SnV⁻ centre emission. The inset shows a PL image of the site and its nearby surroundings. **b** Spectrum at 4.2 K of the same site, showing the $\gamma$ and $\delta$ optical transitions typical of a SnV⁻ centre. The inset shows the basic energy level structure and the optical transitions of the SnV⁻ centre. The off-resonant excitation is indicated in green. **c** Histogram showing the background-corrected second-order auto-correlation measurement of PL emitted from the same site as in panels (**a**, **b**). The red curve corresponds to a three-level model $g^{(2)}(\delta t) = 1 - (1+\alpha)e^{-\frac{|\delta t|}{\tau_1}} + \alpha e^{-\frac{|\delta t|}{\tau_2}}$, with fitted values $\tau_1 = 1.4 \pm 1.5$ ns and $\tau_2 = 9.7$ ns. The red shaded region represents one standard error. **d** Polarisation dependence of the integrated emission of all zero-phonon lines (ZPLs, black), as well as the $\gamma$ (blue) and $\delta$ (orange) optical transitions as a function of analysing polariser angle. The integrated ZPL intensity is normalised to its maximum value, while the $\gamma$ and $\delta$ lines are normalised relative to the maximum intensity of the $\gamma$ line.

linearly (scaling exponent $0.89 \pm 0.15$) with implantation dose – the increase in fluorescence from the Type II Sn (scaling exponent $0.31 \pm 0.05$) and SnV⁻ (scaling exponent $0.59 \pm 0.27$) with ion dose are sub-linear, suggesting that larger doses lead to lower fluorescence quantum yields post-activation or a wider range of defect structures being formed. Strikingly, multiple sharp peaks are observed in the SnV⁻ region in Fig. 1b (bottom panel), suggestive of a population of distinct SnV⁻ colour centres subject to different local strain environments.

**Laser-activated single tin-vacancy centre**

By applying laser annealing to arrays containing an average of 10 ions per site, we now demonstrate the capability to laser-activate single implanted SnV⁻ centres. All characterisation were conducted post-laser annealing. Figure 2a shows a room temperature PL spectrum (processed to remove Raman scattering) from one of the array sites, with the associated PL image in the inset. The spectrum reveals a remarkably clean ZPL at 619 nm and its accompanying phonon side-band (PSB) extending to 750 nm[31], characteristic of a single SnV⁻ colour centre. At cryogenic temperature (4.2 K), the ZPL splits into two narrow peaks, characteristic of the $\gamma$ and $\delta$ optical transitions, as shown in Fig. 2b, with the corresponding energy levels illustrated in the accompanying level diagram[10]. The splitting between the $\gamma$ and $\delta$ lines is 1.7 THz, suggesting that the SnV⁻ centre is located in a highly strained environment[44].

To confirm that the activated SnV⁻ centre corresponds to a single emitter, we performed a Hanbury Brown-Twiss intensity autocorrelation measurement[45]. Figure 2c shows the background-corrected[46] second-order autocorrelation function $g^{(2)}(\delta t)$, revealing a dip at $\delta t = 0$ with a $g^{(2)}(0)$ value of $0.1 \pm 0.13$, which is characteristic of a single photon emitter (see Methods section). Fitting an analytic function for a three-level system[31] to the $g^{(2)}(\delta t)$ histogram reveals an excited state lifetime of $\sim$1.4 ± 1.5 ns. This value is shorter than the 5 –7 ns lifetimes reported previously for SnV⁻[10,30]. The measured lifetime is likely an effective lifetime, incorporating both radiative and non-radiative contributions. The reduction in lifetime may arise from implantation-induced damage, the presence of nearby defects, or the close proximity of the SnV⁻ centre to the diamond surface (20 nm depth). These factors can lead to enhanced non-radiative decay[30] and emission delays[47]. A further contributing factor might be the influence of high-power off-resonant excitation[48,49].

Fluorescence polarimetry (see Methods section), reveals the expected polarisation dependence for the $\gamma$ and $\delta$ optical transitions of the SnV⁻ centre[19,28–30], see Fig. 2d. This behaviour originates from the electronic structure and symmetry of the SnV⁻ centre in diamond. Group-IV colour centres (e.g., SnV⁻, SiV⁻, GeV⁻, and PbV⁻) are characterised by a split-vacancy configuration with $D_{3d}$ symmetry[17,50,51]. This symmetry imposes constraints on the electronic wavefunctions, leading to well-defined selection rules for optical transitions, with the

optical dipole moment oriented along specific crystallographic axes (<111>). As a result, each optical transition exhibits a specific polarisation dependence depending on the involved orbitals and the orientation of the defect relative to the optical axis, along which the emission is detected.

Notably, the SnV⁻ centres created by laser annealing demonstrated remarkable stability under extensive investigation, even when subjected to high-power off-resonant excitation. These SnV⁻ centres thus exhibit the same stability as expected for group-IV defects created via thermal annealing[10,28–32,42].

### The Type II Sn defect complex

The appearance of a Sn-related defect complex with ZPL in the 595 nm spectral region—which we refer to as 'Type II Sn'—has been reported previously[31,41,42] but a detailed study has yet to be performed. Here we show the creation of single Type II Sn defects and analyse them in more detail, comparing them with the appearance of single SnV⁻ colour centres. Extended laser annealing (see Methods section) of several arrays with a range of implanted ion dosages per site reveals the formation of sharp fluorescence peaks at both 595 nm and 620 nm, indicating the creation of small numbers of Type II Sn and SnV⁻ colour centres. Figure 3a shows the resulting distribution of wavelengths for the dominant (narrowest and most intense) spectral features at room temperature for activated sites. Interestingly, the Type II Sn centres at 595 nm exhibit a much narrower wavelength distribution, with a standard deviation of 1.10 nm, than the SnV⁻ centres at 620 nm, with a standard deviation of 4.77 nm. This suggests differing strain susceptibilities and defect structures.

Figure 3b, c, and d show high-resolution spectra of the Type II Sn photoluminescence recorded at cryogenic and room temperatures, displaying a clear ZPL and red-shifted PSB extending up to 720 nm. The inset of Fig. 3b, acquired with a higher resolution diffraction grating, shows a detailed view of the ZPL that resolves four distinct optical transitions reminiscent of those observed in the SnV⁻ centre. Moreover, in contrast to the SnV⁻ centre, which typically exhibits a ground state (GS) splitting on the order of 850 GHz, this Type II Sn centre was found to have a much smaller GS splitting of 380 GHz. Franck-Condon analysis (see Methods section)[52,53] of the PSB provides a good fit based on single phonon coupling spectra extending to the 165 meV longitudinal optical phonon energy of diamond, and yield a Huang-Rhys factor of 1.70, significantly larger than the value of 0.89 for SnV⁻ [54].

Figure 3e shows an HBT measurement performed on a Type II Sn centre, yielding a $g^{(2)}(0)$ value of 0.3 ± 0.12, confirming single-photon emission, and a fitted lifetime of -2.2 ± 0.3 ns. The polarisation dependence of the optical transitions of the Type II Sn centre is shown in Fig. 3f. The integrated ZPL emission (black) of the Type II Sn centre exhibits no net polarisation dependence. However, closer inspection of the individual ZPLs (inset, Fig. 3b) reveals that the bright optical transitions C and D exhibit a pronounced polarisation dependence, analogous to the $\gamma$ transition of the SnV⁻ centre. Notably, their polarisation responses are orthogonal to each other. Furthermore, it is evident that the D transition has a very different polarisation dependence compared to the $\delta$ transition of the SnV⁻ centre. For the Type II Sn centre, the polarisation dependence of both C and D is similar to that of the SiV⁻ centre[55], an emitter with weaker spin-orbit coupling and GS splitting.

The observations above—the lower inhomogeneous broadening, the different Huang-Rhys factor, the lower ground state splitting and the different polarisation properties—strongly suggest that the Type II Sn centre is an altogether different Sn complex to SnV⁻, and not another optically active charge state of same defect. This is consistent with theoretical predictions which predict the ZPL of the optically-active neutral tin-vacancy centre to be at 681 nm[50].

Figure 3g and h present yield statistics for these two types of emitters. Figure 3g displays a PL map obtained after 5 min of laser annealing of a previously untreated section of an array with an implantation dosage of 10 ions per site. The sites where only Type II Sn emission is observed are circled in red, while sites with only SnV⁻ emission are circled in orange. Figure 3h shows a histogram of $g^{(2)}(0)$ values measured across all sites circled in Fig. 3g (region 1). Additionally, Fig. 3h also shows statistics for another similar laser activated region in the same array (region 2). This histogram indicates that of the sites in the array that show fluorescence, most are single photon emitters and that all five SnV⁻-active sites are single photon emitters. While the total activation yield of implanted ions is still low (based on $g^{(2)}(0)$ values, 43 emitters out of an estimated 320 implanted ions or 13%), this relatively short laser anneal (which due to the low duty cycle of the pulse train in the annealing laser and rapid thermal diffusion in diamond equates to less than a second of energy transfer to the lattice) suggests that higher yields are possible, and the result clearly demonstrates the potential for site selective feedback.

### Dynamic switching between single SnV⁻ and Type II Sn colour centres

Combining the laser annealing technique with PL spectroscopy facilitates the monitoring of dynamic processes at the single defect level. We select a previously untreated region of implanted array sites (with a mean dosage of 10 ions) to showcase this capability. Given the integration time of approximately a minute required for spectral measurements of single defects at ambient conditions, we adopted an iterative protocol alternating between laser annealing for 1 min and the capture of PL maps for the identification of discrete emitters. Figure 4a, b, and c each present a time-ordered sequence of PL spectra (without any background subtraction) for three selected single implantation sites, respectively, showing the evolution of the emission over several anneal steps.

Figure 4a shows a sequence for a site where the spectrum is initially dominated by the second-order Raman signature of diamond, followed from 1 min. to 3 min. of annealing by the appearance of Type II Sn photoluminescence, and then between 4 min. and 5 min. a progressive reduction in Type II Sn emission and appearance of SnV⁻ emission. In the site shown in Fig. 4b, the opposite process is observed: 1 min of annealing results in strong SnV⁻ photoluminescence which switches to Type II Sn photoluminescence after 4 min of laser treatment. Figure 4c describes yet another sequence of events for a site where SnV⁻ emission appears after 1 min of annealing and survives for over 8 min of annealing, albeit with some modest variation in centre wavelength and intensity (see Supplementary Note 5). After 10 min, however, the SnV⁻ emission disappears altogether, leaving only the Raman scattering signal. An HBT measurement of the SnV⁻ emission observed in Fig. 4a is displayed in Fig. 4d and reveals a $g^{(2)}(0)$ value of 0.26, indicating the creation of a single colour centre.

Simultaneously with the spectral monitoring, the switching dynamics of SnV⁻ were probed with higher time resolution by monitoring the total photoluminescence intensity within the 615– 625 nm emission window using a single-photon avalanche diode (SPAD). This method also provides faster, real-time feedback control, allowing annealing to be halted as soon as SnV⁻ centres are activated or deactivated, similar to that used previously in the deterministic creation of NV centres[38]. Figure 4e shows a SPAD trace during the 4–5 min interval of Fig. 4a. After 40 s of laser annealing, a sudden increase in SPAD counts indicates the activation of the SnV⁻ centre. Figure 4f similarly illustrates the sudden deactivation of the SnV⁻ centre during the 3–4 min interval of Fig. 4b (see Supplementary Note 5).

## Discussion

The fast, localised activation of fluorescence reported in Fig. 1 suggests that it results from the direct optical excitation of pre-existing defects at the implantation sites. The high-energy Sn ion implantation process displaces some carbon atoms from their lattice sites, thereby generating both carbon vacancies and self-interstitials ($C_i$)[56]. The absence

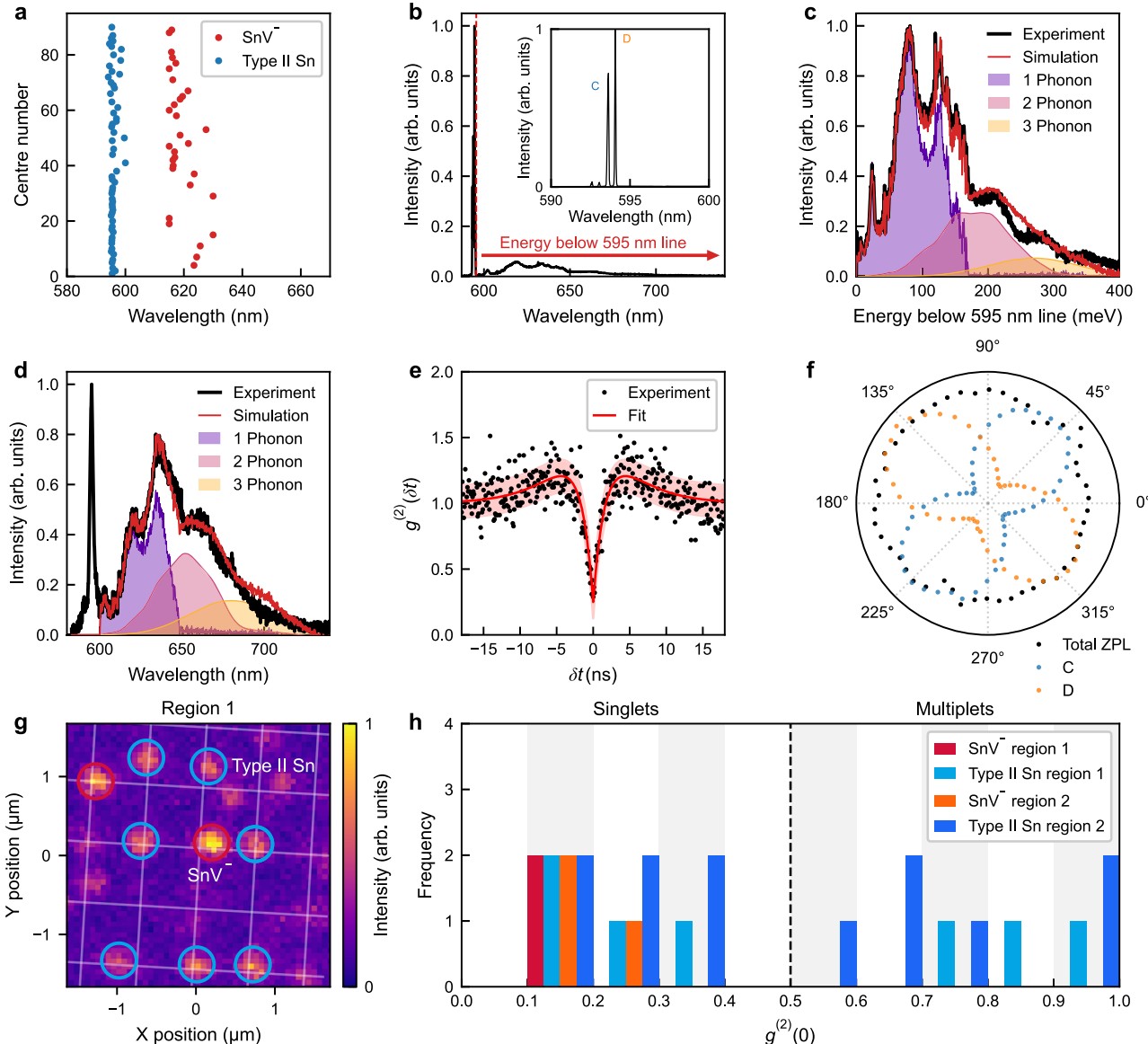

**Fig. 3 | Study of Type II tin (Sn) defect complexes. a** Zero-phonon line (ZPL) distribution of 90 sites activated by laser annealing. **b** Spectrum of a single Type II Sn emitter at a temperature of 4.2 K. The emitter has sharp ZPLs at 595 nm and a phonon sideband (PSB) extending to 720 nm. The inset shows a high resolution spectrum of the ZPLs, where four distinct optical transitions are observed (see also Supplementary Note 4). The two most prominent ones are denoted as optical transitions C and D. **c** Analysis of the PSB at 4.2 K. It is decomposed into multi-phonon contributions. **d** Room temperature spectrum of the same Type II Sn emitter and analysis of the PSB. **e** Second-order autocorrelation measurement with background correction and corresponding fit for the single Type II Sn emitter shown in (**b–d**). A clear dip is observed with $g^{(2)}(0) = 0.3 \pm 0.13$. The red shaded region represents one standard error. **f** Polarisation dependence of the integrated emission of all ZPLs (black), and the C (blue) and D (orange) optical transitions for a single Type II Sn defect at 4.2 K. The integrated ZPL intensity is normalised to its maximum value, while the C and D lines are normalised relative to the maximum intensity of line D. **g** Photoluminescence image of a $3 \, \mu m \times 3 \, \mu m$ region of an array with a dosage of 10 ions/site, taken after 5 min of laser annealing. Nine stable emitting sites are identified, with negatively-charged tin vacancy ($SnV^-$) centres circled in orange and Type II Sn centres circled in red. **h** Histogram of $g^{(2)}(0)$ values for activated stable emitters. It shows the results for two identically sized regions 1 (shown in **g**) and 2 of the same array. Alternating grey and white vertical bands indicate the histogram bins, which have a width of 0.1.

of GR1 fluorescence in the post-implantation spectra suggests that these vacancies are predominantly in the negatively charged (ND1) state, which is not optically active under green excitation and does not emit in the measured wavelength range[57]. While unexpected for electronic grade diamond[58–60], the formation of stable ND1 defects by the ion implantation process may be attributed to band bending effects near the surface[61], resulting from the relatively shallow implantation depth of ~20 nm. The preliminary laser treatment then converts the charge state of some of these ND1s to GR1s[43,62–65], resulting in the emergence of the corresponding luminescence feature centred at 740 nm.

The appearance of Type II Sn and $SnV^-$ photoluminescence on short time scales during laser annealing may also be related to charge state modification, although this is not the hypothesis we deem most plausible based on our observations. Since we observe laser annealing induced switching between these two different types of defects (Type II Sn and $SnV^-$), it is evident that the laser annealing process induces diffusion and thus causes a partial reconfiguration of the crystal lattice.

The complex microscopic mechanisms underlying this process can be outlined as follows. The femtosecond laser pulses trigger both linear and nonlinear absorption mechanisms at the implantation sites[34,39], creating free charge carriers and excitons via multi-photon

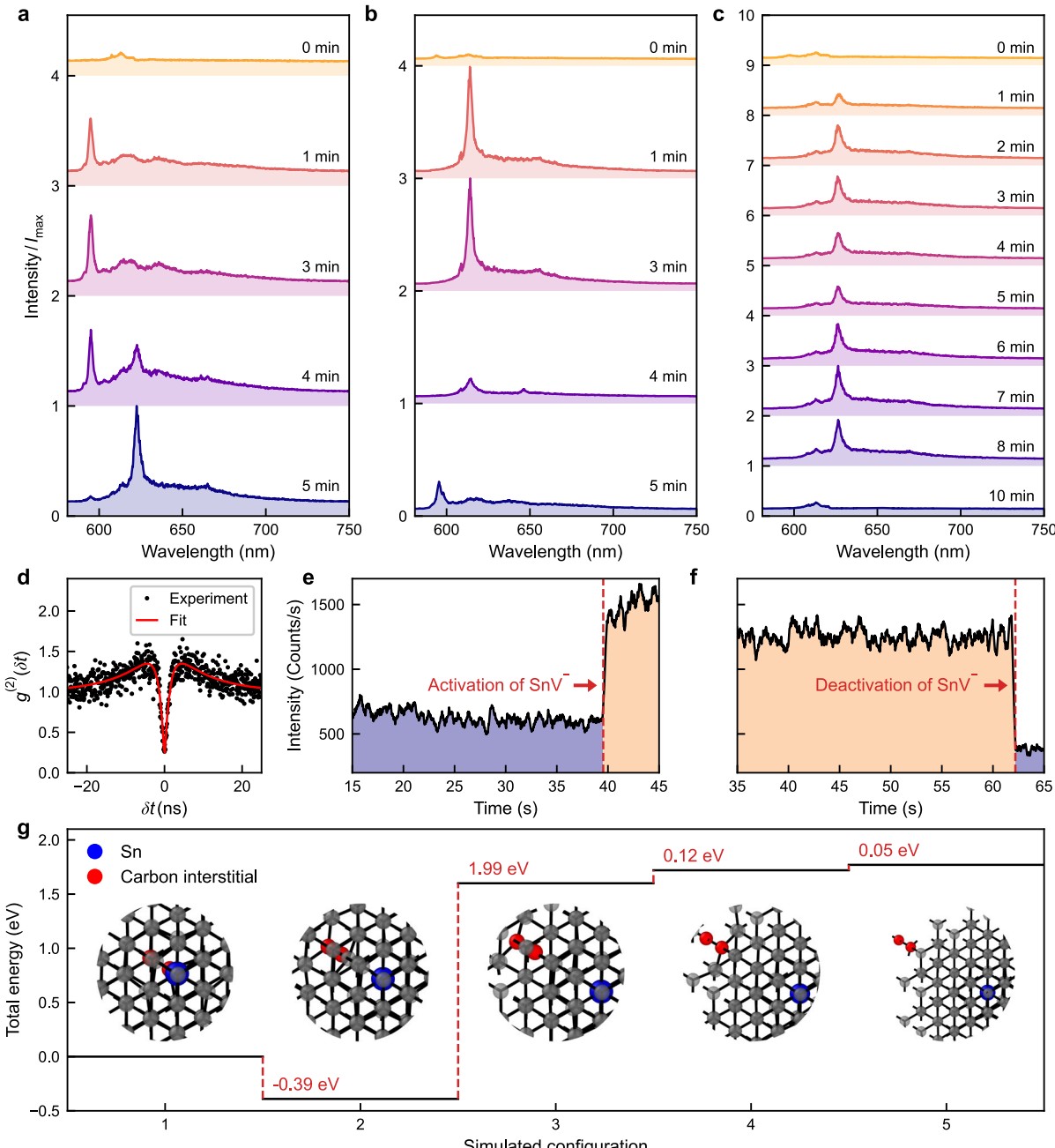

**Fig. 4 | Live spectral monitoring of the temporal evolution of single tin vacancy (SnV⁻) centres during laser annealing.** In subfigures (**a**–**c**) spectra progress from top to bottom, indicating increasing laser annealing time. Within each subfigure, the spectra are normalised to the highest occurring intensity ($I_{max}$). **a** Formation of a single SnV⁻ centre, starting from bare diamond background emission. The spectral evolution shows: the initial formation of a Type II Sn centre, the emergence of SnV⁻ centre emission peaks, and the final SnV⁻ centre spectrum. **b** Formation of a single Type II Sn centre, the spectrum transitions from a Type II Sn centre (weak emission) to a clear SnV⁻ centre, terminating to a Type II Sn centre upon further laser annealing. **c** Deactivation of a SnV⁻ centre. Starting with a Type II Sn centre (weak emission), progresses to a clear SnV⁻ centre. Further laser annealing first leads to modification of the SnV⁻ emission, then deactivation into a bare diamond background emission. **d** Second-order autocorrelation measurement of the SnV⁻ centre formed in (**a**). **e, f** Single photon avalanche detector (SPAD) monitoring during laser annealing with a collection window of 615–625 nm to emphasise the signal from SnV⁻ centres; showing a case of activation and deactivation of SnV⁻ centres. **g** Density functional theory simulations of a negatively charged tin vacancy-carbon self-interstitials (SnV⁻-$C_i$) complex with varying separation between the SnV⁻ and $C_i$. The horizontal axis represents number of a lattice sites separation between the two defects. The two carbon atoms comprising the self-interstitial along [100] are indicated in red.

band-to-band transitions and defect ionisation[34,66–68]. Initially, these free charge carriers and excitons exhibit non-thermal distributions, which then rapidly thermalise through carrier-carrier scattering to a temperature much higher than the lattice on a timescale of a few hundred femtoseconds[69]. The energy stored in these electronic excitations is then partially transferred to the diamond lattice through

carrier-phonon scattering[67–70] and other processes such as non-radiative recombination of charge carriers and excitons[67,71]. The resulting non-thermal phonon distributions will eventually thermalise via scattering processes[69,72], leading to localised lattice heating[67,68,70]. Here, it is interesting to point out that non-thermal phonon distributions could lead to significantly larger diffusion rates compared to

thermal distributions[73]. With increasing energy in the vibrational modes of the crystal lattice, diffusion processes of lattice defects get activated which allows the formation and reconfiguration of colour centres. Since we do not observe thermally induced graphitisation, which sets in at a temperature of 1050 °C[74], it can be inferred that the maximum temperature within the focal volume remains well below this value. Consequently, the thermal load is relatively low and does not produce the high-temperature conditions characteristic of conventional thermal annealing[30]. Simultaneously, the energy contained in the electronic and phononic subsystems diffuses over an area of several $\mu$m in diameter within a few nanoseconds[70]. This is in agreement with our observation that the effects of the extended laser treatment are delocalised to several $\mu$m beyond the diffraction-limited laser spot (see Supplementary Note 3). Through further heat diffusion, the lattice temperature then rapidly cools to ambient conditions on a timescale of several nanoseconds, as a result of the very high thermal conductivity of diamond[70,75]. The speed of this cooling process prevents cumulative heating since our annealing pulses have a temporal spacing of $1\,\mu$s (see Methods section). The above points outline some of the microscopic processes that likely contribute to defect migration during laser annealing. In particular, carbon interstitials in diamond are highly mobile[76–78] and may diffuse short distances as part of the activation process.

Regarding the formation of the 595 nm emission peak and SnV$^-$ centres, previous research has shown that ~40% of implanted Sn ions already settle into a split vacancy configuration[41], necessarily resulting in two $C_i$ defects being present in the near vicinity. An alternate hypothesis is therefore that the resultant SnV-2$C_i$ complex is optically inactive, but that the preliminary laser treatment is sufficient to mobilise one of the $C_i$s, and could leave behind an optically active SnV-$C_i$ complex—the Type II Sn with a feature at 595 nm. Importantly, subsequent extended laser treatment further facilitates the mobilisation of the remaining $C_i$, which is more strongly bound to the SnV defect, allowing it to migrate away from the SnV defect and switching on the SnV$^-$ photoluminescence.

To investigate aspects of this hypothesis we performed density functional theory (DFT) calculations of the energy of a single $C_i$ near to an SnV$^-$ centre. The calculations reveal that a single $C_i$ can form stable complexes with SnV$^-$ without undergoing recombination. As can be seen in Fig. 4g, we find that the energy of the SnV$^-$ centre complexed with a $C_i$ one or two lattice sites away is around 2 eV lower than that of configurations where the carbon interstitial is either more distant from or directly adjacent (but not bound) to the Sn$^-$V centre.

The emerging picture of laser annealing mobilising $C_i$ defects to create different defect complexes is consistent with previous work on nitrogen-vacancy defects[79] and with other aspects of the experimental results presented here. The larger inhomogeneous distribution of spectral lines from SnV$^-$ compared to Type II Sn, shown in Fig. 3a, would be a natural result of the varying strain field experienced by the SnV$^-$ defect as a result of the range of positions that the dissociated $C_i$ could occupy in the nearby lattice. Continued annealing leads to a clearer SnV$^-$ spectrum, as the $C_i$s diffuse away from the SnV$^-$ centres, and creates a more stable environment that favours consistent emission characteristics. However, further annealing can also reintroduce the $C_i$s into the vicinity of the SnV$^-$ centres. In some cases, a $C_i$ will once again bind to the SnV$^-$ centre, producing Type II Sn emission or, if two $C_i$s migrate close enough, deactivate the SnV$^-$ centres entirely (as seen in Fig. 4c).

If correct, our model may also shed light on the low yield of SnV$^-$ centres in high purity diamond following ion implantation, despite the high initial formation efficiency of split-vacancy configurations[28,41]. During the thermal annealing process, a significant proportion of SnV$^-$ centres would likely remain bound to surrounding carbon interstitials, leading to the frequent formation of stable SnV-$C_i$ complexes emitting at 595 nm. Only if sufficient energy is provided to overcome the SnV-$C_i$

binding energy, by e.g. a 2100 °C anneal (requiring high pressure), a large portion of SnV-$C_i$ will remain, as corroborated by previous work[31].

In this study, we combined site-selective, low dose ion implantation and subsequent femtosecond laser annealing to activate group-IV colour centres in electronic-grade diamond. Our approach allows for the activation of both ensemble and single SnV$^-$ centres, depending upon the ion implantation dose. We also presented a thorough investigation on the distinct Type II Sn defect complex which is prevalent during the lattice reconfiguration of diamond implanted with Sn ions.

Our femtosecond laser annealing method offers a rapid and spatially selective approach to annealing under ambient conditions, eliminating the need for extreme high-pressure and high-temperature environments. Furthermore, this technique enables real-time spectral analysis and PL feedback control, offering insights into defect dynamics, such as the observed interplay between single SnV$^-$ centres and Type II Sn configurations. This feedback on various defect creation pathways can facilitate on-demand activation with enhanced yields. Hence, by integrating low dose ion implantation with in-situ monitoring, deterministic activation of single group-IV defect centres is achieved. Note that similar laser annealing of implanted Si ions can also be used to create isolated SiV$^-$ centres (see Supplementary Note 6). Further research into the fabrication parameters—such as femtosecond laser pulse energy, wavelength, repetition rate, pulse duration, and number of pulses - could be very valuable to further optimise this laser annealing process and lead to a more comprehensive understanding of the underlying physical mechanisms. Additionally, microscopic simulations of the laser pulse absorption and subsequent energy diffusion could provide an intuitive physical picture for the involved temperature dynamics. Looking ahead, the technique offers potential for the activation of deterministically implanted single ions, and for annealing with inline monitoring at cryogenic temperatures to enable in-situ tuning of a single defect's fine structure.

## Methods
### Ion Implantation
The diamond substrates were implanted using the Ionoptika Q-One Platform for Nanoscale Advanced Materials Engineering (P-NAME, University of Manchester, UK[25]), which has a maximum anode voltage of 25 kV. The isotopic mass resolution of the ion implantation system enables the selection of the $^{117}$Sn isotope, with choice of an ionisation state of 2+ providing an energy of 50 keV, leading to an implantation depth of ~20 nm (as simulated using Stopping Range of Ions in Matter, SRIM[80]). The alignment markers, designed to be the vertices of a square, were created with 25 keV bismuth ions.

The implantations were conducted in a Poissonian mode. The ion beam current was kept fixed, and the pulse width of the electrostatic beam blanking was varied in order to obtain a desired average number of ions per spot (with the variance governed by Poissonian statistics).

The ion beam was electrostatically blanked between each implantation site, ensuring the only regions to be implanted were those chosen by the implantation design. The high vacuum of the sample chamber ($10^{-8}$ mBar) ensured no contamination during the implantation process.

### Laser activation process
The frequency-doubled output of an ytterbium-doped laser (Spectra Physics Spirit) was used to deliver a train of pulses with duration of 400 fs at a repetition rate of 1 MHz and a central wavelength of 520 nm. The laser beam was expanded and directed onto a Meadowlark liquid crystal spatial light modulator (SLM), which was in turn imaged in 4f configuration onto the back aperture of an Olympus $50 \times 0.95$ NA objective lens. The SLM gave fine control over position and shape of the focal intensity distribution of the laser and was used to eliminate any system aberration.

For the preliminary laser treatment, the laser focus was uniformly scanned across ion implantation regions with varying dosages. A 6 $\mu$m by 6 $\mu$m area within each array was raster scanned at a speed of 200 nm/s, and a laser pulse energy of 1.7 nJ. Given a diffraction limited laser focal beam diameter of 330 nm, this corresponds to a laser fluence of 2 J cm$^{-2}$, and an effective dose at each point in the raster scan of $1.6 \times 10^6$ pulses. We have chosen the laser fluence to be below the threshold of graphitisation or ablation of the diamond surface[81].

For the extended laser annealing, the laser pulses irradiated the diamond surface with a fixed focal position. The laser focus was placed approximately at the centre of an implantation region that we intended to study. The duration of the laser exposure was determined based on live PL feedback. The pulse energy used for this process was reduced to 1 nJ, corresponding to a laser fluence of 1.2 J cm$^{-2}$, and the laser was focused 1 $\mu$m below the diamond surface to avoid potential graphitisation and ablation during this extended exposure.

## Experimental confocal setup

Room temperature and cryogenic PL imaging and spectroscopy were conducted using a home-built scanning confocal microscope integrated with the fs laser processing system described earlier. At room temperature, PL images and spectra were obtained using a 532 nm continuous wave (CW) laser (Cobalt), operating at a power of 1 mW as the excitation source. The PL was collected through a high numerical aperture (NA = 0.95) air objective lens, shared with the femtosecond laser activation beam. A tailored 582 nm long pass filter was applied when taking PL images to block the first order Raman emission of the diamond. Integrated PL measurements were made using a single photon avalanche detector (Excelitas), while spectral data was acquired with a 300 mm spectrograph (SpectraPRo HRS-300) and low noise camera (PIXIS 100 from Princeton Instruments).

For cryogenic PL mapping, the sample was mounted in a Montana S-50 closed-cycle liquid helium cryostat, integrated with a customised beam-scanning confocal microscope. The sample was mounted upon a 4 mm$^2$ piece of thermal grade diamond using Apiezon thermal N-grease and silver dag. Additionally, the thermal grade diamond was thermally coupled to a cold finger with thermal N-grease. The cold-finger was connected to a three stage, open-loop piezoelectric actuator for navigation. Sample imaging was acquired with a Zeiss EC Epi-plan 100x, 0.85 NA, 0.87 mm working distance objective lens. The objective lens was mounted within the cryostat housing under vacuum, and held at room temperature with an internal heater.

## Phonon autocorrelation measurements and data fitting

The background-corrected[46] second-order autocorrelation function, $g^{(2)}(\delta t)$ is defined by

$$g^{(2)}(\delta t) = \frac{\langle I(t)I(t+\delta t)\rangle}{\langle I(t)\rangle^2} \tag{1}$$

where $I(t + \delta t)$ is PL intensity at a time difference of $\delta t$. The fitting function for a three-level system[30] is defined by

$$g^{(2)}(\tau) = 1 - (1+\alpha)e^{-\frac{|\tau|}{\tau_1}} + \alpha e^{-\frac{|\tau|}{\tau_2}}. \tag{2}$$

where $\alpha$, $\tau_1$ and $\tau_2$ are all fitting parameters that relate to interlevel rate constants and the fitted $\tau_1$ can be used to estimate the excited state lifetime of the emitter. The red sold lines in the autocorrelation plots in the main text are a least-squares fit using this equation.

## Polarisation measurements

To characterise the polarisation properties of the light emitted from the colour centres, we used a motorised half-wave plate (HWP) in the detection beam path. This HWP can be precisely rotated to vary the polarisation state of the transmitted light. The HWP introduces a phase shift between the orthogonal components of the light, effectively rotating its polarisation direction by twice the angle of the waveplate's rotation. Following the HWP, a linear polariser was positioned to transmit only the component of the electric field aligned with its transmission axis, thereby blocking the orthogonal component. The linearly polarised light was then coupled into a single-photon avalanche diode (SPAD) for intensity measurement. By rotating the HWP and recording the transmitted light intensity, we analysed the polarisation response of the SnV$^-$ centres using Malus's law, which relates the intensity of transmitted light to the angle between the light's polarisation direction and the polariser's axis.

The PL emission intensities for fluorescence polarimetry were obtained by integrating the spectrum over the zero-phonon line (ZPL) wavelength range. For Type II Sn defects, the integration range was 593–595 nm for the full ZPL, 593.5–593.8 nm for the optical transition C and 594.0–594.3 nm for the optical transition D discussed in the main text. For the SnV$^-$ centre, the integration was performed over 619–622 nm for the $\gamma$ transition and 622–625 nm for the $\delta$ transition.

## Franck Condon analysis

To model the PL emission spectrum of Type II Sn, we applied the Franck-Condon principle[52,53]. The emission intensity as a function of photon energy, $I(E)$, is expressed as:

$$I(E) \propto E^3 \sum_{n=1}^{\infty} I_n(E_v)|M_{0n}|^2, \tag{3}$$

where $E$ represents the photon energy, $I_n(E_v)$ is the lineshape corresponding to the creation of $n$ phonons with vibrational energy $E_v$, and $|M_{0n}|^2$ is the squared overlap integral. This expression captures the sum of all possible phonon sidebands, each weighted by its Franck-Condon factor and modulated by the photon density of states.

The Franck-Condon factor quantifies the probability of emission into the $n$-phonon vibrational level of the ground electronic state, and is given by:

$$|M_{0n}|^2 \propto \frac{S^n}{n!}, \tag{4}$$

where $S$ is the Huang-Rhys factor. This factor reflects the strength of the coupling between electronic states and vibrational modes, which in turn influences the relative intensity of the phonon sidebands.

The lineshape for the transition involving the creation of $n$ phonons, $I_n(E_v)$, is determined recursively through the following convolution of the single-phonon lineshape $I_1(E_v)$:

$$I_n(E_v) = \int_0^{E_{max}} I_1(x)I_{n-1}(E_v - x)\,dx. \tag{5}$$

Here, $E_v$ denotes the vibrational energy of the lattice immediately following the optical transition, and $E_{max}$ is the maximum energy of the $n$-phonon distribution. This recursive convolution process aggregates the contributions of successive phonon interactions, forming the overall lineshape for the phonon sideband. The emission spectrum $I(E)$ is a cumulative outcome of the weighted contributions from each phonon sideband. Each term in the sum reflects the probability of a specific phonon transition, as modulated by the Franck-Condon factors and the photon density of states.

## DFT model for SnV$^-$-C$_i$ complex

Simulations were performed using a $3 \times 3 \times 3$ supercell containing a negatively charged tin-vacancy complex (SnV$^-$) and a carbon interstitial. When introducing one interstitial carbon into the diamond lattice, it undergoes energetic optimisation to form a split-$\langle 100 \rangle$ interstitial carbon pair. Various configurations were explored with the

interstitial placed at different lattice sites relative to the SnV⁻ centre. These simulations indicate the presence of a stable configuration for the Type II Sn centre in which the interstitial does not recombine with the vacancies. The samples containing SnV and $C_i$ are constructed through CrystalMaker[82] and then applied to DFT for geometry optimisation based on the CASTEP code[83]. The Perdew-Burke-Ernzerhof (PBE) exchange-correlation functional[84] is utilised with ultrasoft pseudopotentials. The cutoff energy is set to 900 eV and the Monkhorst-Pack (MP) grid spacing is specified to 0.05.

## Data availability
The data generated in this study have been deposited in the Apollo database[85].

## Code availability
The code for plotting the data in the main manuscript and for the heat simulation in the supplementary information is included in the Apollo database[85]. The code for the density-functional theory simulations is available from the corresponding authors upon request.

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

## Acknowledgements

This work was supported in part by the Quantum Computing and Simulation Hub (EP/T001062/1) through the Partnership Resource Fund PRF-09-I-06 (D.A.G., A.T.), by the Henry Royce Institute for Advanced Materials funded through EPSRC grants EP/R00661X/1, EP/S019367/1, EP/P025021/1 and EP/P025498/1, by the EPSRC through grants EP/R025576/1 and EP/V001914/1, and by capital investment by the University of Manchester (M.C., M.Ad., R.J.C.), by the EPSRC through grant EP/W025256/1 (P.S.S.), and by the ERC Advanced Grant PEDESTAL (884745) (C.P.M., M.At.). G.S.J. and C.P.M. acknowledge support from the EPSRC DTP. D.A.G. acknowledges support from a Royal Society University Research Fellowship. A.C.F. acknowledges support from EU Quantum Flagships, ERC Grants Hetero2D, and EPSRC Grants EP/K01711X/1, EP/K017144/1, EP/N010345/1, EP/L016087/1, EP/V000055/1, EP/X015742/1, EP/Y035453/1. We thank Element Six Ltd and Matthew Markham for providing electronic grade diamond for this project and IonOptika Ltd for discussions related to the Sn source.

## Author contributions

X.C., A.T., G.S.J., J.E.B. and C.P.M. conducted the experiments and data analysis. G.C. performed the DFT calculations. O.B. processed the diamond sample. M.C. and M.Ad. performed the ion implantation. All authors contributed to the discussions of the data. The manuscript was written by X.C., A.T., J.M.S., P.S.S., and D.A.G., with contributions from all authors. D.A.G., J.M.S., and P.S.S. conceived the experiments. D.A.G., J.M.S., P.S.S., R.J.C., A.C.F., and M.At. supervised the work.

## Competing interests

The authors declare no competing interests.
