## [Transparent Peer Review file · Nature Communications]

Laser Activation of Single Group-IV Colour Centres in Diamond

Corresponding Author: Professor Dorian Gangloff

Version 0:

Reviewer comments:

Reviewer #1

(Remarks to the Author)

This manuscript reported a two-step process for single-defect positions and activation of SnV⁻ color centers, in diamonds using site-controlled ion implantation and femtosecond laser annealing. The activation dynamics of the SnV⁻ color centers are in situ evaluated using PL spectra monitoring. They demonstrated site-selective creation and modification of highly ordered array of single SnV⁻ centers with sub-50 nm resolution, providing a promising route to scale up quantum technology. The in-situ spectral monitoring offers new insights into dynamics of defect structures and charging states of color centers, which is of great significance in the development of quantum technology. The manuscript is well written and flow smoothly, I think it is acceptable for publication in Nature Communications after addressing the following issues:

1. In this study, the lifetime of the SnV⁻ center is estimated to be approximately 1.4-1.5 ns, which is notably shorter than previously reported values. The author attributes this to rapid radiative decay; however, they might overlook the potential impact of nearby defects or impurities, which could lead to charge state fluctuations or non-radiative recombination, thereby shortening the lifetime. This scenario could indeed be plausible, especially when considering cross-talk effects from the ion implantation array. Additionally, the lifetime of the SnV⁻ should vary with temperature due to energy losses via phonon interactions. The author should consider including these factors in their analysis.
2. The conventional activation route typically involves high temperatures (1200°C at low pressure or 2100°C at high pressure). To prevent significant damage to diamond surfaces, the authors used femtosecond (fs) laser post-annealing to activate color centers in diamonds. However, femtosecond lasers are often described as a "cold" process because of the extremely rapid laser-material interactions. Consequently, the annealing mechanism of fs laser annealing might differ significantly from that of traditional thermal annealing. The authors should provide further evidence to clarify the mechanism by which fs laser annealing activates color centers.
3. The authors selected a femtosecond laser with a wavelength of 520 nm for annealing. However, the effectiveness and mechanism of fs laser annealing might vary if different wavelengths, such as 350 nm or 1040 nm, are used. Each wavelength could interact differently with the diamond lattice, potentially affecting the absorption, penetration depth, and the type of defects created or modified during the annealing process. Please clarify this.
4. The γ transition components in the SnV⁻ emission are highly polarized, which is quite intriguing. Understanding the factors that influence the polarization of color centers is vital for applications where control over polarization is essential, such as in quantum information processing, where polarization can encode quantum information. The author is advised to offer more insight into the mechanisms behind the formation of this polarized emission.

Reviewer #2

(Remarks to the Author)

The manuscript reports a technique that combines ion implantation and subsequent femtosecond laser annealing to activate group-IV SnV centers in electronic-grade diamond. In comparison with previous methods, this technique presents a novel approach for generating quantum emitters and monitoring the dynamic processes at the single-defect level. I consider this work to be both highly innovative and practical, and I recommend it for publication in Nature Communications. However, prior to publication, I believe the authors should address the following questions:

- a) The authors demonstrate that subsequent femtosecond laser irradiation can enhance the brightness of the injected region. However, the underlying physical mechanism of the laser's role in this process remains unclear. Is the extended laser irradiation locally heating the sample, or is there a more complex interaction between the laser pulses and the injected

defects? Clarifying the specific role of the laser is critical.

b) This could be further explored by varying additional laser parameters, such as reducing the laser repetition rate (to minimize thermal effects) or controlling the number of laser pulses (potentially down to a single pulse). These modifications would help to isolate and better understand the influence of thermal effects.

c) Is it feasible to incorporate thermal field simulations for the laser irradiation to provide a more detailed and intuitive representation of the thermal distribution within the interaction area? The results from such simulations could be compared to the experimental data for a more comprehensive analysis.

d) Will the optical defects induced by the laser annealing in the injection region remain stable over time? This aspect warrants further investigation.

Version 1:

Reviewer comments:

Reviewer #1

(Remarks to the Author)

The authors have sufficiently addressed the reviewer comments regarding the color center lifetime, femtosecond laser activation mechanism, laser wavelength selection, and polarized defect emission. In my opinion, the manuscript is suitable for publication in Nature Communications

Reviewer #2

(Remarks to the Author)

The authors have adequately addressed my previous questions. Their responses are clear and well-supported. I appreciate the revisions made, which enhance the clarity and overall quality of the manuscript.

Response Letter

REVIEWER 1

This manuscript reported a two-step process for single-defect positions and activation of SnV⁻ color centers, in diamonds using site-controlled ion implantation and femtosecond laser annealing. The activation dynamics of the SnV⁻ color centers are in situ evaluated using PL spectra monitoring. They demonstrated site-selective creation and modification of highly ordered array of single SnV⁻ centers with sub-50 nm resolution, providing a promising route to scale up quantum technology. The in-situ spectral monitoring offers new insights into dynamics of defect structures and charging states of color centers, which is of great significance in the development of quantum technology. The manuscript is well written and flow smoothly, I think it is acceptable for publication in Nature Communications after addressing the following issues:

Comment 1

In this study, the lifetime of the SnV⁻ center is estimated to be approximately 1.4-1.5 ns, which is notably shorter than previously reported values. The author attributes this to rapid radiative decay; however, they might overlook the potential impact of nearby defects or impurities, which could lead to charge state fluctuations or non-radiative recombination, thereby shortening the lifetime. This scenario could indeed be plausible, especially when considering cross-talk effects from the ion implantation array. Additionally, the lifetime of the SnV⁻ should vary with temperature due to energy losses via phonon interactions. The author should consider including these factors in their analysis.

We thank the reviewer for the insightful comments regarding the measured lifetime of the SnV⁻ centres in our study. We appreciate the opportunity to clarify and expand upon our analysis of the observed lifetime of 1.4–1.5 ns.

As we mentioned in our manuscript in section B, paragraph 2, the short lifetime we observed might be attributable to the close proximity of the emitter to the diamond surface (20 nm depth), leading to rapid surface-related non-radiative decay, as observed by other research groups [1].

We also agree that crosstalk effects from the ion implantation could play a role. The lattice spacing of the implantation sites was 0.78 μm and there were 10 tin ions implanted per site. While crosstalk between different implantation sites might be possible, we think that the main source of crosstalk, if present, might come from the other tin ions implanted to the same site, as these ions lead to additional defects and impurities in close proximity to our tin vacancy. As the reviewer correctly points out, these nearby defects and impurities could lead to additional non-radiative recombination pathways [2] as well as charge state fluctuations, leading to a reduction in lifetime.

Regarding the temperature dependence of the lifetime, we agree with the reviewer that – in general - excited-state lifetimes are expected to slowly decrease with increasing temperature due to phonon-assisted non-radiative decay processes [3]. However, we would like to highlight that our g_2 measurement was not performed during annealing, but at cryogenic temperatures (4.2 K), as is standard for these emitters [4]. Therefore, the temperature dependence of the excited-state lifetime is less of a concern here.

We would also like to emphasize that the exact value of the lifetime is not central for the purpose of this work. The characteristics of single defects are expected to vary depending on their local environment. What is important is that the created defect is correctly identified and is a single photon emitter. Given that all other characteristics match expectations (room temperature spectrum [1, 5], cryogenic spectrum [1, 5], polarization properties of spectral lines [6], $g_2(0) < 0.5$ [1, 5]), we are confident that the created color center is a single negatively charged tin vacancy.

In response to the reviewer's feedback, we have added more detail to section B, paragraph 2 in the revised manuscript.

“This value is shorter than the 5 – 7 ns lifetimes reported previously for SnV⁻ [1, 4]. The measured lifetime is likely an effective lifetime, incorporating both radiative and non-radiative contributions. The reduction in lifetime may arise from implantation-induced damage, the presence of nearby defects, or the close proximity of the SnV⁻ centre to the diamond surface (20 nm depth). These factors can lead to enhanced non-radiative decay [1] and emission delays [7]. A further contributing factor might be the influence of high-power off-resonant excitation [8, 9].”

This section now also explicitly acknowledges the potential impact of nearby defects and impurities likely induced by the additional tin ions implanted into the same spot.

Comment 2

The conventional activation route typically involves high temperatures (1200°C at low pressure or 2100°C at high pressure). To prevent significant damage to diamond surfaces, the authors used femtosecond (fs) laser post-annealing to activate color centers in diamonds. However, femtosecond lasers are often described as a "cold" process because of the extremely rapid laser-material interactions. Consequently, the annealing mechanism of fs laser annealing might differ significantly from that of traditional thermal annealing. The authors should provide further evidence to clarify the mechanism by which fs laser annealing activates color centers.

We thank the reviewer for highlighting the important distinction between femtosecond (fs) laser annealing and conventional thermal annealing. We agree that the mechanism of fs laser annealing might indeed differ significantly from that of traditional thermal annealing.

In traditional thermal annealing, the phonon population is always described by a thermal distribution. When the diamond lattice is hot, occasionally, one atom or defect in the lattice receives a large enough amount of energy from these lattice vibrations and jumps to a neighbouring lattice site, leading to diffusion. This diffusion can then lead to the creation of color centers and to a healing of the lattice damage created by ion implantation.

The reviewer correctly points out that in the field of laser micromachining, femtosecond micromachining is often referred to as a "cold" process. This is because the duration of these femtosecond pulses is shorter than the characteristic timescale of the electron-lattice interaction (\sim few ps), which leads to minimal heat dissipation into the surrounding bulk material during laser ablation [10]. For diamond, the mechanism for this laser ablation is as follows. Through local energy deposition (for example multiphoton absorption), the sp^3 -hybridised bonds of diamond can be converted to sp^2 -hybridised bonds, a process called graphitisation [10]. If the amount of deposited energy is high enough, these graphitised layers can then sublime, leading to ablation or etching of the diamond surface [10]. More details on this can be found in the review article on femtosecond laser micromachining by Ali et al. [10].

We emphasize that we do not operate in this high pulse energy regime. For our laser annealing experiments, we chose our pulse energies such that there was no measurable damage (graphitisation or ablation) to the diamond surface on the timescale of the experiment. Both graphitisation and ablation can be - if present - readily identified with photoluminescence measurements [11, 12] and optical microscopy during fabrication.

More specifically, for diamond surfaces, previous research has found an excitation wavelength independent graphitisation threshold fluence of $2 - 3 \text{ J cm}^{-2}$ for pulses with a duration of 100 fs [13]. For our laser annealing experiments, we used fluences of $\leq 2 \text{ J cm}^{-2}$ (maximum pulse energy 1.7 nJ, focus diameter 330 nm) together with a four times longer pulse duration of 400 fs to make sure that we stay below this threshold.

In the lower pulse energy regime that we used, the expected dynamics are very different from graphitisation and ablation. Since these ultrafast many-body dynamics are very complex [10, 14–17], it cannot be stated with certainty what the exact mechanism is for laser annealing, as likely many processes contribute simultaneously. Nevertheless, we will outline some of the involved processes and the associated timescales below.

- Our femtosecond lasers pulses generate free charge carriers and excitons through multi-photon band-to-band transitions and defect ionisation [12, 16–18].
- Initially, these free charge carriers and excitons are described by non-thermal distributions and then rapidly thermalise through carrier-carrier scattering to a temperature much higher than the lattice on the timescale of a few hundred femtoseconds [19].
- The energy stored in this excited electronic system is then transferred to the diamond lattice through scattering with optical and acoustic phonons on the timescale of tens of picoseconds [17, 19–21]. Other possible pathways that can lead to localised energy deposition into the lattice include non-radiative recombination of charge-carriers and excitons [15, 17].
- Initially, the resulting phonon distribution will be non-thermal and then eventually thermalise via scattering processes [14, 19]. Here, it is interesting to point out that non-equilibrium phonon distributions might lead to vastly higher diffusion rates compared to thermal distributions [22]. With increasing energy in the vibrational modes of the crystal lattice, the atoms in the focal spot start to vibrate more and more, eventually leading to lattice defect diffusion and color center creation.

- The surface of diamond begins to graphitise at steady-state temperatures of 1050 °C [23]. Since we do not observe such thermally induced graphitisation (even for exposure times > 12 h), the peak temperatures reached during laser annealing are likely below this value. It is therefore unlikely that we reach the high temperature conditions characteristic of conventional thermal annealing (e.g. 1200 °C at low pressure or up to 2100 °C at high pressure).
- On a timescale of several tens of nanoseconds, the energy contained in the electronic and phononic subsystems then diffuses over an area of several μm in diameter [21].
- On the same timescale, the lattice temperature quickly cools to ambient conditions [21, 24], due to the very large thermal conductivity of diamond. This prevents cumulative heating, since our laser annealing pulse train has a repetition rate of 1 MHz, corresponding to a temporal spacing of 1 μs . More details on this can be found in our response to comment 3 of reviewer 2.

In our laser annealing experiments, we observed the creation of tin vacancies and tin related complexes on very short timescales compared to conventional thermal annealing, which typically requires hours. This becomes especially clear when considering that the time of effective heating per pulse is unlikely to be longer than 10 ns, see our response to comment 3 of reviewer 2 and the work by Kononenko et al. [21]. In our case, the effective annealing time per 1 minute of laser annealing (1 MHz repetition rate) - which was shown to produce color centers in Fig.4 - is thus only on the order of 600 ms. We therefore hypothesise that - in addition to standard thermally activated diffusion processes - the creation of our color centers is in part driven by non-equilibrium processes [22].

In response to the reviewer's comment and for the benefit of the reader we have used the above points to substantially expand the discussion section in the revised manuscript. It now clarifies in detail which microscopic processes might contribute to color center creation via femtosecond laser annealing.

"The appearance of Type II Sn and SnV^- photoluminescence on short time scales during laser annealing may also be related to charge state modification, although this is not the hypothesis we deem most plausible based on our observations. Since we observe laser annealing induced switching between these two different types of defects (Type II Sn and SnV^-), it is evident that the laser annealing process induces diffusion and thus causes a partial reconfiguration of the crystal lattice.

The complex microscopic mechanisms underlying this process can be outlined as follows. The femtosecond laser pulses trigger both linear and nonlinear absorption mechanisms at the implantation sites [12, 18], creating free charge carriers and excitons via multi-photon band-to-band transitions and defect ionisation [12, 16, 17, 20]. Initially, these free charge carriers and excitons exhibit non-thermal distributions, which then rapidly thermalise through carrier-carrier scattering to a temperature much higher than the lattice on a timescale of a few hundred femtoseconds [19]. The energy stored in these electronic excitations is then partially transferred to the diamond lattice through carrier-phonon scattering [17, 19–21] and other processes such as non-radiative recombination of charge carriers and excitons [15, 17]. The resulting non-thermal phonon distributions will eventually thermalise via scattering processes [14, 19], leading to localised lattice heating [17, 20, 21]. Here, it is interesting to point out that non-thermal phonon distributions could lead to significantly larger diffusion rates compared to thermal distributions [22]. With increasing energy in the vibrational modes of the crystal lattice, diffusion processes of lattice defects get activated which allows the formation and reconfiguration of colour centres. Since we do not observe thermally induced graphitisation, which sets in at a steady-state temperature of 1050 °C [23], it can be inferred that the maximum temperature within the focal volume remains well below this value. Consequently, the thermal load is relatively low and does not produce the high-temperature conditions characteristic of conventional thermal annealing [1]. Simultaneously, the energy contained in the electronic and phononic subsystems diffuses over an area of several μm in diameter within a few nanoseconds [21]. This is in agreement with our observation that the effects of the extended laser treatment are delocalised to several μm beyond the diffraction-limited laser spot (SI section C). Through further heat diffusion, the lattice temperature then rapidly cools to ambient conditions on a timescale of several tens of nanoseconds, as a result of the very high thermal conductivity of diamond [21, 24]. The speed of this cooling process prevents cumulative heating since our annealing pulses have a temporal spacing of 1 μs (see Methods section). The above points outline some of the microscopic processes that likely contribute to defect migration during laser annealing. In particular, carbon interstitials in diamond are highly mobile [25–27] and may diffuse short distances as part of the activation process."

We have further added the following sentence to the Methods section to clarify how we chose the laser fluences that we used for annealing.

"We have chosen the laser fluence to be below the threshold of graphitisation or ablation of the diamond sur-

face [13].”

For the benefit of the reader, we now also directly specify the laser fluence that we used for laser annealing instead of specifying the pulse energy and the focal beam diameter separately.

Comment 3

The authors selected a femtosecond laser with a wavelength of 520 nm for annealing. However, the effectiveness and mechanism of fs laser annealing might vary if different wavelengths, such as 350 nm or 1040 nm, are used. Each wavelength could interact differently with the diamond lattice, potentially affecting the absorption, penetration depth, and the type of defects created or modified during the annealing process. Please clarify this.

We thank the reviewer for raising this important point regarding the potential influence of laser wavelength on the fs-laser-annealing process. In this study, we are constrained by our experimental setup, which is optimized for operation at 520 nm. While other wavelengths (e.g., 350 nm or 1040 nm) could offer interesting insights, exploring them would require significant reconfiguration of the system, which is beyond the scope of the current work. We emphasize that the 520 nm wavelength used here has proven effective to demonstrate the fabrication of single SnV and type II Sn centres, which is the focus of this work. We acknowledge that other studies have explored fabrication using different wavelengths [28], and we agree that investigating wavelength-dependent effects - as, for example, done by Kononenko et al. [13] - could be valuable for future research.

In response to the reviewer’s comment we have updated the conclusion section of the revised manuscript. There, we now highlight that future research into fabrication parameters could help to further optimize the fabrication process.

”Further research into the fabrication parameters - such as femtosecond laser pulse energy, wavelength, repetition rate, pulse duration, and number of pulses - could be very valuable to further optimise this laser annealing process and lead to a more comprehensive understanding of the underlying physical mechanisms.”

Comment 4

The γ transition components in the SnV⁻ emission are highly polarized, which is quite intriguing. Understanding the factors that influence the polarization of color centers is vital for applications where control over polarization is essential, such as in quantum information processing, where polarization can encode quantum information. The author is advised to offer more insight into the mechanisms behind the formation of this polarized emission.

We thank the reviewer for highlighting the intriguing polarisation properties of the γ transition components in the SnV⁻ emission. Indeed, the polarization behaviour of these transitions is a critical aspect for applications such as quantum information processing, where polarisation can be used as a degree of freedom to encode quantum information.

The polarisation properties of the optical transitions of the SnV⁻ centre are well understood based on previous theoretical and experimental work [1, 6, 29–31]. This behaviour originates from the electronic structure and symmetry of the SnV⁻ centre in diamond. Group-IV colour centres (e.g., SnV⁻, SiV⁻, GeV⁻, and PbV⁻) are characterised by a split-vacancy configuration with D_{3d} symmetry [32–34]. This symmetry imposes constraints on the electronic wavefunctions, leading to well-defined selection rules for optical transitions, with the optical dipole moment oriented along specific crystallographic axes ($\langle 111 \rangle$). As a result, each optical transition exhibits a specific polarization dependence depending on the involved orbitals and the orientation of the defect relative to the optical axis, along which the emission is detected.

In response to the reviewer’s comment, we have added additional context and information to the paragraph discussing the polarization dependence of the γ transition.

”Fluorescence polarimetry (see Methods section), reveals the expected polarisation dependence for the γ and δ optical transitions of the SnV⁻ centre [1, 6, 30, 31], see Fig. 2d. This behaviour originates from the electronic structure and symmetry of the SnV⁻ centre in diamond. Group-IV colour centres (e.g., SnV⁻, SiV⁻, GeV⁻, and PbV⁻) are characterised by a split-vacancy configuration with D_{3d} symmetry [32–34]. This symmetry imposes constraints on the electronic wavefunctions, leading to well-defined selection rules for optical transitions, with the optical dipole moment oriented along specific crystallographic axes ($\langle 111 \rangle$). As a result, each optical transition exhibits a specific polarisation dependence depending on the involved orbitals and the orientation of the defect relative to the optical axis, along which the emission is detected.”

To make the difference in polarization properties between the SnV^- and Type II Sn centre clearer, we have further added more detailed data to both Fig. 2d and Fig 3b,f. For both types of emitters, we now show the polarisation dependence of the integrated emission of all ZPLs, as well as of the two brightest optical transitions. From these plots it is now directly evident that the D transition of the Type II Sn centre has different polarisation properties compared to the δ transition of SnV^- .

Moreover, from Fig. 3b, the extracted ground state (GS) splitting for the Type II Sn centre is 380 GHz, which is much smaller than the minimal GS splitting for SnV^- (850 GHz). It is further interesting to note that the polarisation characteristics of the Type II Sn centre are similar to that of the SiV^- centre [35], an emitter with a much smaller spin-orbit coupling and ground state (GS) splitting.

For the benefit of the reader we have added the above information to section C.

Section C, paragraph 2:

”The inset of Fig. 3b, acquired with a higher resolution diffraction grating, shows a detailed view of the ZPL that resolves four distinct optical transitions reminiscent of those observed in the SnV^- centre. Moreover, in contrast to the SnV^- centre, which typically exhibits a ground state (GS) splitting on the order of 850 GHz, this Type II Sn centre was found to have a much smaller GS splitting of 380 GHz.”

Section C, paragraph 3 and 4:

”The polarisation dependence of the optical transitions of the Type II Sn centre is shown in Fig. 3f. The integrated ZPL emission (black) of the Type II Sn centre exhibits no net polarisation dependence. However, closer inspection of the individual ZPLs (inset, Fig. 3b) reveals that the bright optical transitions C and D exhibit a pronounced polarisation dependence, analogous to the γ transition of the SnV^- centre. Notably, their polarisation responses are orthogonal to each other. Furthermore, it is evident that the D transition has a very different polarisation dependence compared to the δ transition of the SnV^- centre. For the Type II Sn centre, the polarisation dependence of both C and D is similar to that of the SiV^- centre [35], an emitter with weaker spin-orbit coupling and GS splitting.

The observations above – the lower inhomogeneous broadening, the different Huang-Rhys factor, the lower ground state splitting and the different polarisation properties – strongly suggest that the Type II Sn centre is an altogether different Sn complex to SnV^- , and not another optically active charge state of same defect. This is consistent with theoretical predictions which predict the ZPL of the optically-active neutral tin-vacancy centre to be at 681 nm [32]. ”

REVIEWER 2

The manuscript reports a technique that combines ion implantation and subsequent femtosecond laser annealing to activate group-IV SnV centers in electronic-grade diamond. In comparison with previous methods, this technique presents a novel approach for generating quantum emitters and monitoring the dynamic processes at the single-defect level. I consider this work to be both highly innovative and practical, and I recommend it for publication in Nature Communications. However, prior to publication, I believe the authors should address the following questions:

Comment 1

The authors demonstrate that subsequent femtosecond laser irradiation can enhance the brightness of the injected region. However, the underlying physical mechanism of the laser’s role in this process remains unclear. Is the extended laser irradiation locally heating the sample, or is there a more complex interaction between the laser pulses and the injected defects? Clarifying the specific role of the laser is critical.

We thank the reviewer for the comment and the opportunity to clarify the underlying physical mechanism of femtosecond laser annealing. First of all, we would like to point out that the ultrafast many-body dynamics involved in this process are extremely complex [10, 14–17, 19]. This means that, currently, neither we nor other research groups [18] can say with certainty what the exact mechanisms are. Clarifying this would require substantial further research. Nevertheless, from literature [10, 14–19] and the observations made in our experiments, there is an emerging underlying physical picture. We currently think that it is mainly a combination of local heating and non-equilibrium processes that drive diffusion processes in the diamond lattice and that are therefore responsible for annealing. We have summarized our current understanding in response to comment 2 of reviewer 1, who also was very curious about the underlying physical mechanism of laser annealing. We therefore kindly point to this response for details.

In response to the reviewer's comment, we have substantially expanded the discussion section to clarify the laser's role in the annealing process.

Comment 2

This could be further explored by varying additional laser parameters, such as reducing the laser repetition rate (to minimize thermal effects) or controlling the number of laser pulses (potentially down to a single pulse). These modifications would help to isolate and better understand the influence of thermal effects.

We thank the reviewer for the suggestion regarding the exploration of laser annealing parameters. Varying and optimizing these parameters will certainly be useful for future research, however, it is beyond the scope of the present work, where our focus is a first proof-of-concept demonstration that SnV^- and Type II Sn centres can be created and transformed into one another by laser annealing.

That said, we agree that cumulative thermal effects are certainly an important consideration. In our response to comment 3 below, we show that cumulative heating effects are not expected when operating at repetition rates of 1 MHz. This is possible since diamond is an excellent heat conductor.

In response to the reviewer's comment we added a sentence into the conclusion section of the revised manuscript, where we now highlight that further research into the fabrication parameters could help to optimize the fabrication process and lead to a more comprehensive understanding of the underlying mechanisms.

"Further research into the fabrication parameters - such as femtosecond laser pulse energy, wavelength, repetition rate, pulse duration, and number of pulses - could be very valuable to further optimise this laser annealing process and lead to a more comprehensive understanding of the underlying physical mechanisms."

Comment 3

Is it feasible to incorporate thermal field simulations for the laser irradiation to provide a more detailed and intuitive representation of the thermal distribution within the interaction area? The results from such simulations could be compared to the experimental data for a more comprehensive analysis.

We thank the reviewer for this suggestion. We agree that thermal field simulations would offer an intuitive view of the temperature dynamics during laser irradiation. However, a quantitative simulation that can be directly compared with experimental data would require a full self-consistent many-body simulation of the complex dynamics described in our response to comment 2 of reviewer 1. This is a non-trivial task and would require a separate simulation-focused work with the aim to build a theoretical model for the laser annealing process. Also, for diamond the multiphoton absorption coefficients for femtosecond pulses at 520 nm (2.38 eV) are not well known experimentally. Diamond has a bandgap of ~ 5.5 eV, which requires at least a three-photon process ($3 \cdot 2.38$ eV = 7.41 eV) to bridge it. This means that it is not known what fraction of the pulse energy is absorbed, which would be necessary to properly estimate the energy that is delivered into the focal volume and therefore is available to heat the diamond lattice.

Despite the above limitations, we can use a diffusion equation to get an intuition for the speed at which heat diffusion happens in diamond given certain initial conditions. For a heat profile $T(\mathbf{r}, t)$ given as a function of position \mathbf{r} and time t , the diffusion equation is given by

$$\frac{\partial T(\mathbf{r}, t)}{\partial t} = \alpha \nabla^2 T(\mathbf{r}, t), \quad (1)$$

where $\alpha = k/(\rho c_p)$ is the thermal diffusivity, k the thermal conductivity, ρ the density and c_p the specific heat. For the initial temperature distribution, we assumed a three dimensional gaussian

$$T(\mathbf{r}, 0) = T_b + (T_h - T_b) \exp\left[-\left(\frac{x^2}{2\sigma_x^2} + \frac{y^2}{2\sigma_y^2} + \frac{z^2}{2\sigma_z^2}\right)\right], \quad (2)$$

with standard deviations σ_x , σ_y , σ_z given by the focusing conditions of the laser beam (see Methods). For the initial peak temperature, we chose a value of $T_h = 1050$ °C, motivated by the fact that diamond starts to graphitise at such a temperature under steady-state conditions [23]. The absence of graphitisation in our experiments, despite tests with >12 h of exposure, suggests that our actual peak temperatures remain well below this estimate. The ambient background

temperature was set to $T_b = 25^\circ\text{C}$. The time evolution of the initial temperature distribution can be calculated by convolution with the corresponding Green's function.

In the relevant temperature range, the thermal diffusivity of diamond decreases with increasing lattice temperature. For analytic solutions, we therefore consider two cases, room temperature, and high temperature ($\sim 930^\circ\text{C}$, material parameters for even higher temperatures were not available). Using room temperature parameters yields a diffusivity of $\alpha_{\text{rt}} = 1.22 \cdot 10^{-3} \text{ m}^2/\text{s}$ [36, 37] and overestimates the cooling rate. Using high temperature parameters yields a diffusivity of $\alpha_{\text{hot}} = 0.06 \cdot 10^{-3} \text{ m}^2/\text{s}$ [36, 37] and underestimates the cooling rate. The real temperature dependence is therefore bounded by those two cases, which is a good enough estimate for our purposes here.

The results of this calculation are summarised in Fig. 1 below. Figure 1a shows a top view of the initial lateral temperature distribution. The optical axis is along the z-axis. Figure 1b presents the temperature profiles along the x-axis at successive points in time, showing the diffusion of the temperature distribution. Figure 1c shows the maximum temperature as a function of time for both cases of thermal diffusivity we are considering. From the results it is evident that the heat dissipates within $\sim 10 \text{ ns}$ in either case, two orders of magnitude faster than the time between two consecutive annealing pulses ($1 \mu\text{s}$). This rapid cooling shows that no cumulative heating is expected in our experiments. Figure 1d shows the spread of the temperature distribution within the first nanosecond. It can be seen that the temperature distribution spreads out at most to a few micrometers in diameter before almost cooling down to ambient conditions.

This is consistent with our observations (see Fig. 10), where extended laser annealing affected a region of several micrometers in diameter. Further experimental research has been conducted into heat dissipation in diamond following pulsed femtosecond excitation at near infrared wavelengths [21]. The authors observed heat dissipation over a similar sized region within a few nanoseconds [21], consistent with our results.

It is further interesting to note that the energy contained in the temperature distribution shown in Fig. 1 is 0.14 nJ, which for a 1 nJ input pulse - as we used in our experiments - would correspond to 14 % absorption. In reality, the absorption is likely much lower. While the three-photon absorption coefficient β_3 at 520 nm has not been experimentally determined, measurements indicate that $\beta_{3,520\text{nm}} \ll \beta_{3,400\text{nm}} = 2.3 \cdot 10^{-22} \text{ cm}^3\text{W}^{-2}$ [38]. Using $\beta_{3,400\text{nm}}$ as an upper bound, we can use [38]

$$\frac{\Delta I}{\Delta z} \approx -\beta_{3,400\text{nm}} I_0^3 \quad (3)$$

to estimate the fraction of absorbed pulse energy as

$$\frac{\Delta E}{E_0} = \frac{\Delta I}{I_0} \approx -\beta_{3,400\text{nm}} I_0^2 \Delta z = 16 \%. \quad (4)$$

Here, the effective interaction length $\Delta z = 790 \text{ nm}$ is taken as the confocal parameter, $E_0 = 1 \text{ nJ}$ is the initial pulse energy, $I_0 = 3 \cdot 10^{12} \text{ Wcm}^{-2}$ is the initial peak intensity, and ΔE , ΔI_0 are the respective changes in pulse energy and peak intensity. The peak intensity is given by $I_0 \approx F/\tau$, where $F = 1.2 \text{ Jcm}^{-2}$ is the fluence and $\tau = 400 \text{ fs}$ the pulse duration. For our experimental conditions, the expected fraction of absorbed pulse energy is therefore $\ll 14 - 16 \%$, which means that it is to be expected that the maximum temperatures reached in our experiments are significantly lower than $T_h = 1050^\circ\text{C}$.

In response to the reviewer's comment, we have added the following sentence to the conclusion section:

"Additionally, microscopic simulations of the laser pulse absorption and subsequent energy diffusion could provide an intuitive physical picture for the involved temperature dynamics."

For the benefit of the reader, we have also included the above discussion as a new section in the Supplementary Information.

"For the laser annealing process it is useful to have an estimate for the timescales on which heat diffusion happens in diamond. One limitation of such a simulation is that, for diamond, the multi-photon absorption coefficients for femtosecond laser pulses at 520 nm are not well known experimentally. This means that it is hard to properly estimate the energy that is delivered into the focal volume and therefore available to heat the diamond lattice.

However, despite the above limitation, it is still instructive to use a thermal diffusion equation to get an intuition for the rate at which heat diffusion happens in diamond given certain initial conditions. For a heat profile $T(\mathbf{r}, t)$ given as a

function of position \mathbf{r} and time t , the diffusion equation is given by

$$\frac{\partial T(\mathbf{r}, t)}{\partial t} = \alpha \nabla^2 T(\mathbf{r}, t), \quad (5)$$

where $\alpha = k/(\rho c_p)$ is the thermal diffusivity, k the thermal conductivity, ρ the density and c_p the specific heat. For the initial temperature distribution, we used a three dimensional gaussian

$$T(\mathbf{r}, 0) = T_b + (T_h - T_b) \exp\left[-\left(\frac{x^2}{2\sigma_x^2} + \frac{y^2}{2\sigma_y^2} + \frac{z^2}{2\sigma_z^2}\right)\right], \quad (6)$$

with standard deviations $\sigma_x, \sigma_y, \sigma_z$ given by the focusing conditions of the laser beam (see Methods). For the initial peak temperature, we chose a value of $T_h = 1050^\circ\text{C}$, motivated by the fact that diamond starts to graphitise at such a temperature under steady-state conditions [23]. The absence of graphitisation in our experiments, despite tests with >12 h of exposure, suggests that our actual peak temperatures remain well below this estimate. The ambient background temperature was set to $T_b = 25^\circ\text{C}$. The time evolution of the initial temperature distribution can be calculated by convolution with the corresponding Green's function.

In the relevant temperature range, the thermal diffusivity of diamond decreases with increasing lattice temperature. For analytic solutions, we therefore consider two cases, room temperature, and high temperature ($\sim 930^\circ\text{C}$, material parameters for even higher temperatures were not available). Using room temperature parameters yields a diffusivity of $\alpha_{\text{rt}} = 1.22 \cdot 10^{-3} \text{ m}^2/\text{s}$ [36, 37] and overestimates the cooling rate. Using high temperature parameters yields a diffusivity of $\alpha_{\text{hot}} = 0.06 \cdot 10^{-3} \text{ m}^2/\text{s}$ [36, 37] and underestimates the cooling rate. The real temperature dependence is therefore bounded by those two cases, which provides a good enough estimate for our purposes here.

The results of this calculation are summarised in Fig. 1 below. Figure 1a shows a top view of the initial lateral temperature distribution. The optical axis is along the z-axis. Figure 1b presents the temperature profiles along the x-axis at successive points in time, showing the diffusion of the temperature distribution. Figure 1c shows the maximum temperature as a function of time for both cases of thermal diffusivity we are considering. From the results it is evident that the heat dissipates within ~ 10 ns in either case, two orders of magnitude faster than the time between two consecutive annealing pulses (1 μs). This rapid cooling shows that no cumulative heating is expected in our experiments. Figure 1d shows the spread of the temperature distribution within the first nanosecond. It can be seen that the temperature distribution spreads out at most to a few micrometers in diameter before almost cooling down to ambient conditions.

This is consistent with our observation, see Fig. 10, where extended laser annealing affected a region of several micrometers in diameter. Further experimental research has been conducted into heat dissipation in diamond following pulsed femtosecond excitation at near infrared wavelengths [21]. The authors observed heat dissipation over a similar sized region within a few nanoseconds [21], consistent with our results.

It is further interesting to note that the energy contained in the temperature distribution shown in Fig. 1 is 0.14 nJ, which for a 1 nJ input pulse - as we used in our experiments - would correspond to 14 % absorption. In reality, the absorption is likely much lower. While the three-photon absorption coefficient β_3 at 520 nm has not been experimentally determined, measurements indicate that $\beta_{3,520\text{nm}} \ll \beta_{3,400\text{nm}} = 2.3 \cdot 10^{-22} \text{ cm}^3\text{W}^{-2}$ [38]. Using $\beta_{3,400\text{nm}}$ as an upper bound, we can use [38]

$$\frac{\Delta I}{\Delta z} \approx -\beta_{3,400\text{nm}} I_0^3 \quad (7)$$

to estimate the fraction of absorbed pulse energy as

$$\frac{\Delta E}{E_0} = \frac{\Delta I}{I_0} \approx -\beta_{3,400\text{nm}} I_0^2 \Delta z = 16 \%. \quad (8)$$

Here, the effective interaction length $\Delta z = 790 \text{ nm}$ is taken as the confocal parameter, $E_0 = 1 \text{ nJ}$ is the initial pulse energy, $I_0 = 3 \cdot 10^{12} \text{ Wcm}^{-2}$ is the initial peak intensity, and $\Delta E, \Delta I_0$ are the respective changes in pulse energy and peak intensity. The peak intensity is given by $I_0 \approx F/\tau$, where $F = 1.2 \text{ J cm}^{-2}$ is the fluence and $\tau = 400 \text{ fs}$ the pulse duration. For our experimental conditions, the expected fraction of absorbed pulse energy is therefore $\ll 14 - 16 \%$, which means that it is to be expected that the maximum temperatures reached in our experiments are significantly lower than $T_h = 1050^\circ\text{C}$.

FIG. 1. **Spatiotemporal thermal dynamics in ultrapure diamond.** **a**, Cross-sectional temperature distribution in XY plane immediately after thermal excitation (1050°C initial peak). **b**, X-axis temperature profiles at successive timepoints assuming room-temperature (α_{rt}) diffusivity, demonstrating nanosecond-scale thermal equilibration. **c**, Maximum temperature as a function of time for room-temperature (α_{rt}) and high-temperature (α_{hot}) diffusivity parameters. **d**, Time evolution of the full width at half maximum (FWHM) for both cases.

Based on the above discussion, we now also mention in section A, paragraph 3, that for a repetition rate of 1 MHz, cumulative heating effects are not expected.

”For such a repetition rate, cumulative heating effects are not expected (see SI section A).”

Comment 4

Will the optical defects induced by the laser annealing in the injection region remain stable over time? This aspect warrants further investigation.

We appreciate the reviewer’s comment regarding the long-term stability of optical defects induced by laser annealing. In our experiments, the SnV⁻ centres and Type II Sn-related defects have demonstrated remarkable stability over extended periods. Specifically, the same diamond sample has been subjected to repeated optical characterizations over two years, including numerous cryostat loadings and cleanings, without any observable degradation of these defects. Furthermore, we have illuminated the sample with continuous-wave 532 nm laser light at intensities of up to 30 mW for extended durations (spanning several months), and the optical properties of both SnV⁻ centres and Type II Sn defects have remained unchanged. This is consistent with all experimental work on group-IV colour centres to date.

These observations strongly suggest that the defects generated by our ion implantation and laser annealing process are robust and maintain their optical properties over time under typical experimental conditions.

In response to the reviewer’s remark, we now comment on the stability of the SnV⁻ centers created via laser annealing in section B.

”Notably, the SnV⁻ centres created by laser annealing demonstrated remarkable stability under extensive investigation, even when subjected to high-power off-resonant excitation. These SnV⁻ centres thus exhibit the same stability as

expected for group-IV defects created via thermal annealing [1, 4–6, 30, 39, 40].”

- [1] J. Görlitz, D. Herrmann, G. Thiering, P. Fuchs, M. Gandil, T. Iwasaki, T. Taniguchi, M. Kieschnick, J. Meijer, M. Hatano, A. Gali, and C. Becher, Spectroscopic investigations of negatively charged tin-vacancy centres in diamond, *New Journal of Physics* **22**, 13048 (2020).
- [2] F. A. Inam, A. M. Edmonds, M. J. Steel, and S. Castelletto, Tracking emission rate dynamics of nitrogen vacancy centers in nanodiamonds, *Applied Physics Letters* **102**, 253109 (2013).
- [3] D. K. Bommedi and A. D. Pickel, Temperature-dependent excited state lifetimes of nitrogen vacancy centers in individual nanodiamonds, *Applied Physics Letters* **119**, 254103 (2021).
- [4] M. E. Trushev, B. Pingault, N. H. Wan, M. Gündoğan, L. De Santis, R. Debroux, D. Gangloff, C. Purser, K. C. Chen, M. Walsh, J. J. Rose, J. N. Becker, B. Lienhard, E. Bersin, I. Paradeisanos, G. Wang, D. Lyzwa, A. R.-P. Montblanch, G. Malladi, H. Bakhru, A. C. Ferrari, I. A. Walmsley, M. Atatüre, and D. Englund, Transform-Limited Photons From a Coherent Tin-Vacancy Spin in Diamond, *Phys. Rev. Lett.* **124**, 023602 (2020).
- [5] T. Iwasaki, Y. Miyamoto, T. Taniguchi, P. Siyushev, M. H. Metsch, F. Jelezko, and M. Hatano, Tin-vacancy quantum emitters in diamond, *Phys. Rev. Lett.* **119**, 253601 (2017).
- [6] A. E. Rugar, C. Dory, S. Sun, and J. Vučković, Characterization of optical and spin properties of single tin-vacancy centers in diamond nanopillars, *Phys. Rev. B* **99**, 205417 (2019).
- [7] I. Y. Eremchev, A. O. Tarasevich, M. A. Kniazeva, J. Li, A. V. Naumov, and I. G. Scheblykin, Detection of Single Charge Trapping Defects in Semiconductor Particles by Evaluating Photon Antibunching in Delayed Photoluminescence, *Nano Letters* **23**, 2087 (2023).
- [8] D. V. Regelman, U. Mizrahi, D. Gershoni, E. Ehrenfreund, W. V. Schoenfeld, and P. M. Petroff, Semiconductor quantum dot: A quantum light source of multicolor photons with tunable statistics, *Phys. Rev. Lett.* **87**, 257401 (2001).
- [9] P. Michler, A. Imamoglu, A. Kiraz, C. Becher, M. Mason, P. Carson, G. Strouse, S. Buratto, W. Schoenfeld, and P. Petroff, Nonclassical Radiation from a Single Quantum Dot, *physica status solidi (b)* **229**, 399 (2002).
- [10] B. Ali, I. V. Litvinyuk, and M. Rybachuk, Femtosecond laser micromachining of diamond: Current research status, applications and challenges, *Carbon* **179**, 209 (2021).
- [11] V. V. Kononenko, I. I. Vlasov, E. V. Zavedeev, A. A. Khomich, and V. I. Konov, Correlation between surface etching and NV centre generation in laser-irradiated diamond, *Applied Physics A* **124**, 226 (2018).
- [12] Y.-C. Chen, P. S. Salter, S. Knauer, L. Weng, A. C. Frangeskou, C. J. Stephen, S. N. Ishmael, P. R. Dolan, S. Johnson, B. L. Green, G. W. Morley, M. E. Newton, J. G. Rarity, M. J. Booth, and J. M. Smith, Laser writing of coherent colour centres in diamond, *Nature Photonics* **11**, 77 (2017).
- [13] V. V. Kononenko, V. M. Gololobov, M. S. Komlenok, and V. I. Konov, Nonlinear photooxidation of diamond surface exposed to femtosecond laser pulses, *Laser Physics Letters* **12**, 096101 (2015).
- [14] T. Apostolova, V. Kurylo, and I. Gnilitzkyi, Ultrafast Laser Processing of Diamond Materials: A Review, *Frontiers in Physics* **9**, 10.3389/fphy.2021.650280 (2021).
- [15] S. Mao, F. Quéré, S. Guizard, X. Mao, R. Russo, G. Petite, and P. Martin, Dynamics of femtosecond laser interactions with dielectrics, *Applied Physics A* **79**, 1695 (2004).
- [16] R. R. Gattass and E. Mazur, Femtosecond laser micromachining in transparent materials, *Nature Photonics* **2**, 219 (2008).
- [17] B. Griffiths, A. Kirkpatrick, S. S. Nicley, R. L. Patel, J. M. Zajac, G. W. Morley, M. J. Booth, P. S. Salter, and J. M. Smith, Microscopic processes during ultrafast laser generation of frenkel defects in diamond, *Phys. Rev. B* **104**, 174303 (2021).
- [18] J. Engel, K. Jhuria, D. Polley, T. Lühmann, M. Kuhrke, W. Liu, J. Bokor, T. Schenkel, and R. Wunderlich, Combining femtosecond laser annealing and shallow ion implantation for local color center creation in diamond, *Applied Physics Letters* **122**, 234002 (2023).
- [19] J. Shah, *Ultrafast Spectroscopy of Semiconductors and Semiconductor Nanostructures*, edited by M. Cardona, P. Fulde, K. Von Klitzing, H.-J. Queisser, R. Merlin, and H. Störmer, Springer Series in Solid-State Sciences, Vol. 115 (Springer Berlin Heidelberg, Berlin, Heidelberg, 1999).
- [20] T. Ichii, Y. Hazama, N. Naka, and K. Tanaka, Study of detailed balance between excitons and free carriers in diamond using broadband terahertz time-domain spectroscopy, *Applied Physics Letters* **116**, 231102 (2020).
- [21] V. V. Kononenko, E. V. Zavedeev, M. I. Latushko, and V. I. Konov, Observation of fs laser-induced heat dissipation in diamond bulk, *Laser Physics Letters* **10**, 036003 (2013).
- [22] K. Gordiz, S. Muy, W. G. Zeier, Y. Shao-Horn, and A. Henry, Enhancement of ion diffusion by targeted phonon excitation, *Cell Reports Physical Science* **2**, 100431 (2021).
- [23] F. N. Li, P. C. Zhang, P. F. Zhang, and H. X. Wang, Thermal annealing induced graphite/diamond structure processed by high-voltage hydroxide ion treatments, *Applied Surface Science* **657**, 159753 (2024).
- [24] A. Kirkpatrick, *Optical Engineering of Colour Centres in Diamond*, Ph.D. thesis, University of Oxford, Oxford (2023).
- [25] K. Iakubovskii, I. Kiflawi, K. Johnston, A. Collins, G. Davies, and A. Stesmans, Annealing of vacancies and interstitials in diamond, *Physica B: Condensed Matter Proceedings of the 22nd International Conference on Defects in Semiconductors*, **340–342**, 67 (2003).

- [26] D. J. Twitchen, D. C. Hunt, C. Wade, M. E. Newton, J. M. Baker, T. R. Anthony, and W. F. Banholzer, The production and annealing stages of the self-interstitial (R2) defect in diamond, *Physica B: Condensed Matter* **273–274**, 644 (1999).
- [27] T. Lühmann, J. Meijer, and S. Pezzagna, Charge-Assisted Engineering of Color Centers in Diamond, *physica status solidi (a)* **218**, 2000614 (2021).
- [28] Y.-C. Chen, B. Griffiths, L. Weng, S. S. Nicley, S. N. Ishmael, Y. Lekhai, S. Johnson, C. J. Stephen, B. L. Green, G. W. Morley, M. E. Newton, M. J. Booth, P. S. Salter, and J. M. Smith, Laser writing of individual nitrogen-vacancy defects in diamond with near-unity yield, *Optica* **6**, 662 (2019).
- [29] G. m. H. Thiering and A. Gali, Ab initio magneto-optical spectrum of group-iv vacancy color centers in diamond, *Phys. Rev. X* **8**, 021063 (2018).
- [30] S. D. Tchernij, T. Herzig, J. Forneris, J. Küpper, S. Pezzagna, P. Traina, E. Moreva, I. P. Degiovanni, G. Brida, N. Skukan, M. Genovese, M. Jakšić, J. Meijer, and P. Olivero, Single-Photon-Emitting Optical Centers in Diamond Fabricated upon Sn Implantation, *ACS Photonics* **4**, 2580 (2017).
- [31] X. Guo, A. M. Stramma, Z. Li, W. G. Roth, B. Huang, Y. Jin, R. A. Parker, J. Arjona Martínez, N. Shofer, C. P. Michaels, C. P. Purser, M. H. Appel, E. M. Alexeev, T. Liu, A. C. Ferrari, D. D. Awschalom, N. Deegan, B. Pingault, G. Galli, F. J. Heremans, M. Atatüre, and A. A. High, Microwave-Based Quantum Control and Coherence Protection of Tin-Vacancy Spin Qubits in a Strain-Tuned Diamond-Membrane Heterostructure, *Phys. Rev. X* **13**, 041037 (2023).
- [32] G. Thiering and A. Gali, The $(eg \times eu) \times Eg$ product Jahn–Teller effect in the neutral group-IV vacancy quantum bits in diamond, *npj Comput. Mater.* **5**, 18 (2019).
- [33] C. Hepp, T. Müller, V. Waselowski, J. N. Becker, B. Pingault, H. Sternschulte, D. Steinmüller-Nethl, A. Gali, J. R. Maze, M. Atatüre, and C. Becher, Electronic structure of the silicon vacancy color center in diamond, *Phys. Rev. Lett.* **112**, 036405 (2014).
- [34] L. J. Rogers, K. D. Jahnke, M. W. Doherty, A. Dietrich, L. P. McGuinness, C. Müller, T. Teraji, H. Sumiya, J. Isoya, N. B. Manson, and F. Jelezko, Electronic structure of the negatively charged silicon-vacancy center in diamond, *Physical Review B* **89**, 235101 (2014).
- [35] L. J. Rogers, O. Wang, Y. Liu, L. Antoniuk, C. Osterkamp, V. A. Davydov, V. N. Agafonov, A. B. Filipovski, F. Jelezko, and A. Kubanek, Single $Si-V^-$ centers in low-strain nanodiamonds with bulklike spectral properties and nanomanipulation capabilities, *Phys. Rev. Appl.* **11**, 024073 (2019).
- [36] J. R. Olson, R. O. Pohl, J. W. Vandersande, A. Zoltan, T. R. Anthony, and W. F. Banholzer, Thermal conductivity of diamond between 170 and 1200 k and the isotope effect, *Phys. Rev. B* **47**, 14850 (1993).
- [37] R. R. Reeber and K. Wang, Thermal expansion, molar volume and specific heat of diamond from 0 to 3000k, *Journal of Electronic Materials* **25**, 63 (1996).
- [38] M. Kozák, F. Trojánek, B. Dzurňák, and P. Malý, Two- and three-photon absorption in chemical vapor deposition diamond, *JOSA B* **29**, 1141 (2012).
- [39] A. E. Rugar, H. Lu, C. Dory, S. Sun, P. J. McQuade, Z.-X. Shen, N. A. Melosh, and J. Vučković, Generation of Tin-Vacancy Centers in Diamond via Shallow Ion Implantation and Subsequent Diamond Overgrowth, *Nano Letters* **20**, 1614 (2020).
- [40] E. Corte, S. Sachero, S. Ditalia Tchernij, T. Lühmann, S. Pezzagna, P. Traina, I. P. Degiovanni, E. Moreva, P. Olivero, J. Meijer, M. Genovese, and J. Forneris, Spectral emission dependence of tin-vacancy centers in diamond from thermal processing and chemical functionalization, *Advanced Photonics Research* **3**, 2100148 (2022).